# Completing Missing Annotation: Multi-Agent Debate for Accurate and Scalable Relevance Assessment for IR Benchmarks

**Minjeong Ban,**[*] **Jeonghwan Choi,**[*] **Hyangsuk Min,**[*] **Nicole Hee-Yeon Kim, Minseok Kim, Jae-Gill Lee, Hwanjun Song**[†]
Korea Advanced Institute of Science and Technology
{minjeong.ban, hwani.choi, hyangsuk.min, songhwanjun}@kaist.ac.kr

## Abstract

Information retrieval (IR) evaluation remains challenging due to incomplete IR benchmark datasets that contain unlabeled relevant chunks. While LLMs and LLM-human hybrid strategies reduce costly human effort, they remain prone to LLM overconfidence and ineffective AI-to-human escalation. To address this, we propose DREAM, a multi-round debate-based relevance assessment framework with LLM agents, built on opposing initial stances and iterative reciprocal critique. Through our agreement-based debate, it yields more accurate labeling for certain cases and more reliable AI-to-human escalation for uncertain ones, achieving 95.2% labeling accuracy with only 3.5% human involvement. Using DREAM, we build BRIDGE, a refined benchmark that mitigates evaluation bias and enables fairer retriever comparison by uncovering 29,824 missing relevant chunks. We then re-benchmark IR systems and extend evaluation to RAG, showing that unaddressed holes not only distort retriever rankings but also drive retrieval–generation misalignment. The relevance assessment framework is available at https://github.com/DISL-Lab/DREAM-ICLR-26; and the BRIDGE dataset is available at https://github.com/DISL-Lab/BRIDGE-Benchmark.

## 1 Introduction

Information retrieval (IR) systems aim to retrieve relevant text chunks from large document collections in response to user queries, which is a critical task for many downstream applications (Jiang et al., 2024; Yang et al., 2024; Liu et al., 2025). However, IR evaluation remains challenging primarily due to the limitations of IR benchmark datasets, which rely heavily on costly human annotations of query-chunk relevance. Since only a small subset of text chunks can be labeled, large portions of the corpus remain unjudged, leaving unlabeled yet potentially relevant chunks, so-called "holes" in relevance assessment, which result in potentially misleading evaluation outcomes (Craswell et al., 2020; MacAvaney & Soldaini, 2023; Rassin et al., 2024). These holes undermine the reliability of IR benchmarks by obscuring the true effectiveness of retrieval methods, and the problem becomes even more critical in RAG systems, where unreliable retrieval evaluation hinders understanding and optimization of the retriever-generator interaction (Es et al., 2024; Park et al., 2025).

With the emergence of large language models (LLMs), several efforts have emerged to leverage them to reduce the human cost of constructing IR benchmarks. Recent studies have either fully automated the relevance assessment process (Shivani et al., 2024; Rassin et al., 2024; Park et al., 2025; Rahmani et al., 2025; Ni et al., 2025), or adopted a hybrid strategy in which an LLM handles straightforward cases while humans are engaged only for uncertain instances when the LLM's confidence score is low (Takehi et al., 2025; Li et al., 2025). However, both approaches have inherent limitations due to their strong reliance on a *single* agent serving as the sole judge.

Fully automated pipelines risk propagating the overconfidence of a single model, which often leads to a large number of mislabeled cases (Tian et al., 2025; Sun et al., 2025). In contrast, confidence-

---

[*]Equal contribution.
[†]Corresponding Author.

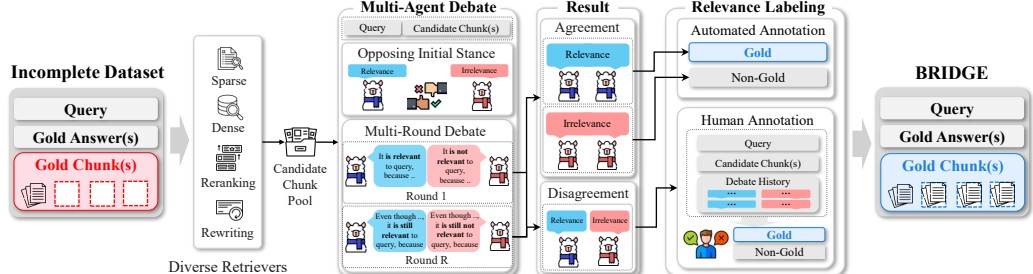

Figure 1: Overview of DREAM pipeline for constructing the BRIDGE benchmark. Multi-agent debate with opposing stances conducts a multi-round process, automatically annotating agreement cases and escalating disagreements to humans with debate history.

based escalation attempts to mitigate failures by sending uncertain cases to human annotators. Yet this strategy has notable limitations: the resulting labels are less accurate than direct human annotation, even for cases the LLM deems certain, and its miscalibrated confidence scores often cause errors by unnecessarily escalating straightforward cases and failing to escalate genuinely ambiguous ones (Kumar et al., 2024; Chhikara, 2025). Consequently, the overall process remains both expensive and error-prone, limiting its effectiveness as a reliable alternative to human annotation.

Moving beyond the constraints of prior single-agent approaches, we suggest a *multi*-agent paradigm (in Figure 1) that harnesses LLMs' true strengths, including diverse reasoning, mutual critique, and collective deliberation. Concretely, we introduce DREAM (Debate-based RElevance Assessment with Multi-agents), a framework that employs *multi-agent adversarial debate* to mitigate holes in IR benchmarks with higher accuracy and lower human cost. Specifically, we initialize two agents with opposing stances, one assuming "relevance" and the other "irrelevance." This design compels agents to surface and challenge conflicting evidence, thereby avoiding premature consensus and mitigating single-perspective bias (Koupaee et al., 2025; Oriol et al., 2025). They then engage in multi-round reciprocal critique, refining evidence and challenging arguments until they either converge to agreement or maintain disagreement.

Unlike recent confidence-based methods (Takehi et al., 2025; Li et al., 2025), our approach leverages *inter-agent agreement* as a direct indicator of reliability. That is, labels are accepted upon agent agreement and escalated when they disagree, avoiding calibration training or threshold tuning. Agreement provides a stronger signal of correctness than a single model's often miscalibrated confidence, while disagreement naturally pinpoints the uncertain cases that require human review. Furthermore, another key distinction of DREAM is its use of the *debate history* as a consistent mechanism across both the multi-agent debate and human annotation stages. During the debate, the history enables agents to reach agreement more efficiently (within two rounds), and when escalation occurs, the same history is provided to humans as a supportive resource, directly assisting them by presenting organized arguments and evidence rather than leaving them to start from scratch. Through this principled and unified design, DREAM achieves a relevance labeling accuracy of 95.2%, surpassing full-scale non-expert labeling while involving humans in only 3.5% of cases.

To validate and showcase our framework, we construct a new BRIDGE benchmark dataset. Instead of creating yet another IR benchmark from scratch, we re-work widely used two IR benchmarks, namely BEIR (Thakur et al., 2021) and RobustQA (Han et al., 2023). By applying our DREAM framework, we systematically identify and fill missing relevant chunks (holes), thereby reducing evaluation bias and producing a refined benchmark that supports fairer comparisons across retrieval systems as well as more faithful assessment of downstream RAG performance. Specifically, in RAG, improvements in retrieval performance do not fully translate into gains in generation performance. The common explanation is that external knowledge often conflicts with the model's internal parametric knowledge (Longpre et al., 2021; Chen et al., 2022; Zhou et al., 2023). While partly valid, we identify *another overlooked factor*: retrieval performance itself has been misestimated due to holes in IR benchmark datasets (see Section 5).

Our main contributions are: (i) we introduce DREAM, a debate-based relevance labeling framework that achieves high accuracy while keeping human effort and cost minimal; (ii) we construct the BRIDGE benchmark by re-labeling BEIR and RobustQA, uncovering 29,824 of missing relevant chunks, equivalent to 428% of the originally annotated 6,976 gold chunks; (iii) we analyze how

holes in existing IR benchmarks distort retrieval performance and rankings, and show that `DREAM` mitigates this bias; and (iv) we present a new insight into the cause of retrieval–generation performance misalignment and address it through our framework.

## 2 RELATED WORK

**Relevance Assessment.** The Cranfield experiment (Cleverdon, 1997), which evaluates IR systems using documents, queries, and human relevance assessments, is a seminal work in relevance assessment. The emergence of LLMs has demonstrated strong capabilities for cost-effectively automating data annotation (de Jesus & Nunes, 2024; Thomas et al., 2024; Upadhyay et al., 2025), replacing expensive human annotation. UMBRELA (Shivani et al., 2024), D-MERIT (Rassin et al., 2024), MIRAGE (Park et al., 2025), and SynDL (Rahmani et al., 2025) have shown that LLMs can assess document relevance as well as humans, while DIRAS (Ni et al., 2025) has demonstrated that fine-tuning smaller LLMs can achieve performance on par with larger models. However, evaluating relevance solely based on LLMs presents limitations, including overconfidence biases and a lack of nuanced contextual understanding (Faggioli et al., 2023; Clarke & Dietz, 2025; Soboroff, 2025).

**Selective Approaches to Relevance Assessment.** Recent work on selective prediction (Varshney et al., 2022; Stengel-Eskin & Durme, 2023; Srinivasan et al., 2024) shows that models can reduce errors by abstaining or escalating uncertain cases rather than committing to unreliable decisions. Building on this idea, AI-human hybrid methods (Xu et al., 2025; Sahitaj et al., 2025), which leverage LLMs for straightforward cases and rely on human intervention for uncertain ones, have been proposed to address inaccuracies in fully automated labeling, such as collaborative annotation (Kim et al., 2024) and verification pipelines (Wang et al., 2024). Yet, there has been limited research on their application to relevance assessment. Beyond the confidence-based method, LARA (Takehi et al., 2025) calibrates LLM confidence into a relevance probability using human-labeled data to mitigate miscalibration and overconfidence. Despite these advances, hybrid pipelines remain limited by single-model accuracy and a heavy reliance on human data. In contrast, our approach mitigates it through opposing opinions and reciprocal critique in multi-round debate, eliminating the need for calibration and the reliance on additional training with human supervision.

**Multi-agent Debate.** Over a single-agent setup, recent studies have explored utilizing multiple agents with distinct roles that collaborate (Chang et al., 2024; Shen et al., 2024). In particular, multi-agent debate has been widely examined, where agents share their answers and rationales, critique one another, and iteratively refine their outputs over multiple rounds (Xiong et al., 2023; Liang et al., 2024a). Such debate-style collaboration has been applied to improve LLM and RAG answer quality (Du et al., 2023; Chen et al., 2024; Khan et al., 2024), to handle knowledge conflicts in RAG (Wang et al., 2025), and to enhance response evaluation (Chan et al., 2023) and data annotation (Tseng et al., 2025). While existing work focuses primarily on AI-AI collaboration, we instead leverage multi-agent debate on AI-Human collaboration for IR relevance assessment. Our framework uses multi-agent debate not only to produce high-quality annotations, but also to determine when to escalate uncertain cases to humans and to improve human judgments through providing debate histories.

**Connecting IR and RAG.** Generation serves as the key downstream task of retrieval, making RAG a natural framework that tightly couples retriever quality (Gao et al., 2023b). Recent studies have shown that holes in IR benchmarks also undermine the reliable evaluation of RAG, as missing relevant chunks create a discrepancy between retrieval and generation performance (Park et al., 2025; Ni et al., 2025). Yet, most RAG benchmark studies continue to focus primarily on joint evaluation of retrieval and generation (Tang & Yang, 2024; Friel et al., 2024; Krishna et al., 2025), with limited attention to holes; even benchmarks such as DIRAS (Ni et al., 2025) and MIRAGE (Park et al., 2025) rely on fully automated LLM-based labeling, leaving annotation reliability unverified.

## 3 AUTOMATED RELEVANCE LABELING FOR IR

We formulate automated relevance labeling as an answer-aware relevance judgment problem, where the task is to decide if a candidate chunk $c$ supports the answer to a given query $q$. Formally, given a target query $q \in \mathcal{Q}$, let $\mathcal{A}(q)$ denote the set of possible answers associated with the query $q$, and $\mathcal{C}(q)$ denote the set of candidate chunks considered for the query $q$. The task is to assign a relevance label

for each tuple $(q, c)$ where $q \in \mathcal{Q}$ and $c \in \mathcal{C}(q)$. Here, a query-chunk pair is labeled as 1 (relevance) if it provides evidential support for at least one answer in $\mathcal{A}(q)$, and 0 (irrelevance) otherwise:

$$f : (q, c) \mapsto y, \quad y = \begin{cases} 1, & \exists\, a \in \mathcal{A}(q) \text{ such that } c \text{ supports } a, \\ 0, & \text{otherwise.} \end{cases} \tag{1}$$

Our goal is to design a labeling function $f$ together with a routing policy, such that (i) certain cases are resolved automatically by $f$, and (ii) genuinely uncertain cases are escalated to humans. The objective is to achieve high labeling accuracy while minimizing the need for human verification.

### 3.1 DREAM: Debate-based Relevance Assessment with Multi-agents

Based on the formulation, we design a labeling function $f$ that leverages *inter-agent agreement* to automate relevance labeling. To realize this principle, we introduce DREAM in Figure 1, a framework that employs multi-round debate between two LLM agents to surface inter-agent (dis)agreement and escalate only genuinely difficult cases. The framework is structured into the following three stages:

**Opposing Stance Initialization.** We begin by initializing LLM agents with *opposing* stances to enforce an explicit divergence of perspectives. Formally, two LLM agents $m_i \in \{m_1, m_2\}$ are instantiated with opposing stances $s_i \in \{s_1, s_2\}$, where $s_1$ is defined as *relevance* ("I think the chunk is relevant to the target query.") and $s_2$ is *irrelevance* ("I think the chunk is not relevant to the target query."). At initialization, $m_1$ is assigned $s_1$, while $m_2$ is assigned $s_2$, ensuring that both perspectives are examined. This forced divergence compels each agent to defend its stance, preventing premature consensus and mitigates single-perspective bias, which also improves factual grounding and deliberative accuracy (Liang et al., 2024b; Koupaee et al., 2025). Importantly, we observe no order dependence in the initial stance assignment even when swapping the initial stances of $m_1$ and $m_2$. Please refer to Appendix B.1 for details.

**Multi-Round Debating with Reciprocal Critique.** Building on opposing stances, DREAM employs a multi-round debate that compels agents to repeatedly confront and critique opposing perspectives, while also allowing them to revise their labeling decisions. During the debate process, each agent comprehensively reviews both its own and the opponent's arguments from previous rounds to generate new reasoning. In each debate round $j$, the two agents $m_i$ are provided a query-chunk pair, its answer set, and the debate history from the previous round $h^j$ as input. Based on this input, each agent critiques the opponent's arguments, extracts evidence sentences from the chunk, and produces a new relevance label $y_i^j$ with its supporting reasoning $r_i^j$.

Specifically, for the first round ($j = 1$), the discussion history is limited to the initial stance $h^1 = \{s_1, s_2\}$, while in subsequent rounds ($j > 1$), the complete discussion history from the previous round $h^j = \{(r_1^{j-1}, y_1^{j-1}), (r_2^{j-1}, y_2^{j-1})\}$ is utilized. After each round concludes, we evaluate inter-agent consensus by verifying whether $y_1^j = y_2^j$. If consensus is reached in any round, the debate terminates with the agreed label, which is then used as the final relevance label; otherwise it proceeds to the next round. The debate continues until either consensus is achieved between the two agents or a predefined maximum number of rounds, $R$, is reached. If the consensus is not ultimately reached, this is considered a genuinely *uncertain* case where the complexity of the relevance assessment exceeds the capability of multi-round debate. Such cases are escalated to human verification. As a result, our designed labeling function is formulated as:

$$\text{DREAM}(q, c) = \begin{cases} y_1^j, & \exists\, j \leq R \text{ s.t. } y_1^j = y_2^j \quad \text{(agreement reached),} \\ \text{Human}(q, c, h^R), & \text{otherwise (persistent disagreement).} \end{cases} \tag{2}$$

Through agent consensus, trivial cases are handled automatically, leaving only uncertain ones for human verification, thereby lowering annotation cost while maintaining accuracy. Refer to the prompts used for the multi-round debate in Appendix A.1.

**Agreement-based Human Escalation.** Persistent disagreement after the maximum rounds $R$ is considered *uncertain*, a case where agents with opposing stances fail to converge. This criterion captures epistemic uncertainty beyond confidence scores, preventing overconfident errors by a single agent and eliminating the need for manually tuned escalation thresholds.

When a case is escalated, annotators receive the query–chunk pair and its answer set, along with the debate history, capturing both agents' reasoning and evidence in the final round, as in Eq. (2). This context helps annotators understand the specific sources of disagreement and the nuanced aspects that make the case challenging, leading to more informed human judgment. Thus, DREAM maintains high accuracy through principled escalation and enhances human judgment by turning escalation into synergy rather than mere fallback, with quantitative gains shown in Section 3.2.4.

## 3.2 COMPARISON WITH EXISTING RELEVANCE LABELING METHODS

We verify the effectiveness of DREAM compared with alternative auto-labeling methods.

**Evaluation Set.** We sample 700 query-answer pairs at random, with 100 pairs from each subset: MS MARCO and NQ from BEIR (Thakur et al., 2021), five domains (Lifestyle, Recreation, Science, Technology, and Writing) from RobustQA (Han et al., 2023). For each query $q$, a candidate chunk $c$ is randomly sampled from a candidate set $\mathcal{C}(q)$ retrieved using 25 different retrievals. As a result, our evaluation set contains 700 queries, where each query $q$ is associated with its answer set $\mathcal{A}(q)$ and the candidate chunk $c$. This corresponds to the input in Eq. (1) of our problem formulation. Note that the evaluation set is a subset of our refined IR benchmark, detailed in Section 4.1.

For this subset, we further collect the ground-truth relevance labels (1: relevance, 0: irrelevance), which are adjudicated by expert human annotators, consisting of graduate-level NLP specialists with complementary academic backgrounds. Refer to Appendix A.2 for details.

**Metrics.** We measure *auto-labeling accuracy* on non-escalated cases by comparing the estimated labels with the expert ground-truth, and we also report the *escalation ratio* to humans. A high value of the former indicates reliable automatic judgments, while a low value of the latter reflects reduced annotation cost and efficient use of human effort. We adopt balanced accuracy (bAcc) as our metric for labeling accuracy. For completeness, we report class-wise recall (relevance and irrelevance) and their average as bAcc, which provides a more robust evaluation than standard accuracy, especially in relevance assessment tasks where irrelevant cases naturally outnumber relevant ones. The escalation ratio denotes the fraction of cases each method deems uncertain and escalates to humans.

**Baselines.** We compare DREAM with two agent-based methods and a reference method that relies entirely on non-expert annotators recruited via Amazon Mechanical Turk (MTurk). The compared methods include: LLMJudge (Shivani et al., 2024; Park et al., 2025), a single-agent method that directly assigns relevance labels without escalation, widely used for automated labeling; LARA (Takehi et al., 2025), a confidence-based escalation method that leverages token probabilities to automatically assign labels and route uncertain cases to humans; and Human-Only, a method that assigns each case to three workers on MTurk, with the final label determined by majority vote (see Section 4.1). For LARA, it requires specifying an escalation ratio, so we adjust it with values in $\{3.5\%, 12.5\%, 25.0\%, 50.0\%\}$, as they represent different levels of human involvement. The 3.5% setting is aligned with DREAM's escalation ratio to ensure fair comparison.

**Implementation.** We use Llama3.3-70B-Instruct (with a temperature of $0.0$) as the LLM agent for all methods, except for Human-Only. Thus, DREAM employs two Llama3.3 models to conduct multi-round debates. The maximum debate rounds $R$ is set to 2 by default, as larger values do not produce meaningful gains (see Section 3.2.2). Details including prompts are in Appendix A.3.

Regarding LARA, it mitigates miscalibration of LLM token probabilities through online training of a logistic regression model, where uncertain (escalated) cases are annotated and used for online training. We apply this setup to our data following the original study (Takehi et al., 2025).

### 3.2.1 MAIN RESULT: ANNOTATION ACCURACY & ESCALATION RATIO

Figure 2 and Table 1 compare the annotation accuracy and escalation ratio of three automatic relevance assessment methods, evaluated on their non-escalated cases, along with one non-expert human-based method Human-Only, which is not a fair comparison but is provided as a reference point. Although accuracy on non-escalated cases can be trivially increased by escalating most cases, the key challenge is achieving high accuracy while keeping escalation ratio low.

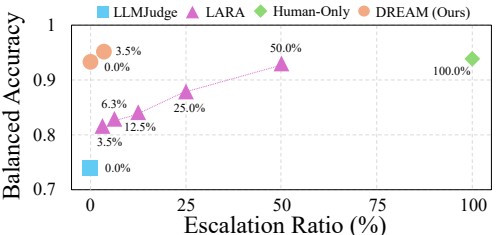

Figure 2: Auto-labeling accuracy and escalation ratio of DREAM compared with two LLMJudge and LARA on non-escalated cases.

Table 1: Comparison of DREAM and two automatic labeling methods on non-escalated cases.

| Method | Class-wise Recall | | bAcc | Escalation Ratio |
|--------|-------------------|---|------|-------------------|
| | Irrelevance | Relevance | | |
| Human-Only | 89.9% | 97.8% | 93.8% | 100.0% |
| LLMJudge | 50.2% | 97.5% | 73.9% | 0.0% |
| LARA | 74.5% | 89.6% | 82.1% | 3.5% |
| | 76.1% | 91.6% | 83.9% | 12.5% |
| | 80.2% | 95.3% | 87.8% | 25.0% |
| | 94.1% | 98.4% | 96.3% | 50.0% |
| DREAM | 91.9% | 98.4% | 95.2% | 3.5% |

The results reveal that DREAM attains automatic relevance judgment performance on par with humans, as evident from its balanced accuracy of 95.2%, surpassing the 93.8% by Human-Only, while requiring escalation in only 3.5% of the cases. The single-agent baseline, LLMJudge, suffers from low recall on irrelevant cases, yielding a much lower balanced accuracy of 73.9% due to the absence of uncertainty-based escalation. Contrarily, LARA outperforms LLMJudge through its confidence-based escalation, achieving higher balanced accuracy with increasing escalation ratio. That is, the confidence-based strategy is partly effective: difficult cases are escalated to humans while the agent handles higher-confidence ones, though misclassifications still occur.

Yet, this escalation proves to be sub-optimal. When restricted to the same 3.5% escalation ratio as DREAM, LARA falls notably short in annotation quality. To reach performance comparable to DREAM, it requires escalating nearly 50.0% of the cases, which demands extensive human involvement and thus lacks scalability. That is, uncertainty estimation in LARA is inaccurate, missing difficult cases while over-escalating easy ones. In contrast, our agreement-based approach with multi-agent debate enables far more precise escalation, achieving higher annotation quality while automatically labeling substantially more cases than LARA with 50% escalation. In particular, unlike LARA, DREAM removes the need for calibration training and manual escalation tuning, making it more practical and scalable for automated relevance assessment.

In addition, DREAM is cost-effective in terms of labeling cost and latency when compared with Human-Only. We conduct a comprehensive cost analysis in Appendix B.2, which confirms that DREAM achieves $200\times$ cheaper and $3.5\times$–$7.0\times$ faster than Human-Only.

### 3.2.2 ABLATION STUDY: DEBATE ROUND AND ADJUDICATION STRATEGY

We conduct an ablation study with DREAM on its two design choices: (i) the number of debate rounds and (ii) the escalation strategy, in which uncertain cases are routed either to human annotators or to an LLM adjudicator, the latter enabling full automation. Regarding (ii), comparing human annotators (via Mturk) with an LLM adjudicator as the final judge is important, as it shows the synergy of AI-human collaboration, which leads to higher annotation quality than fully automated adjudication. See Appendix A.4 for the

Table 2: Overall annotation quality with DREAM for all cases, where escalated cases are adjudicated by either an LLM or humans, and all other cases are labeled automatically by DREAM.

| Method | Adjudicator | Class-wise Recall | | bAcc |
|--------|-------------|-------------------|---|------|
| | | Irrelevance | Relevance | |
| DREAM ($R=1$) | LLM | 82.9% | 97.2% | 90.0% |
| DREAM ($R=2$) | LLM | 90.0% | 96.7% | 93.3% |
| DREAM ($R=3$) | LLM | 90.8% | 95.7% | 93.2% |
| DREAM ($R=2$) | Human | 91.8% | 98.4% | 95.1% |

detailed adjudication setup. Table 2 presents the annotation quality across these settings, illustrating the effect of increasing debate rounds and using different adjudication on overall performance.

First, we increase the maximum round $R$ from 1 to 3 with LLM adjudication, as this setup removes human inconsistency and clearly reveals the impact of debate depth. The results show that performance gains saturate after two rounds, with little improvement beyond. Thus, we set $R = 2$ as the default. Second, human adjudication outperforms LLM adjudication with the same maximum debate round. That is, humans resolve uncertain cases more accurately, thereby supporting our AI-to-human escalation strategy to maximize annotation quality.

We further examine agents' consensus stability across multiple debate rounds, as understanding how spurious consensus emerges or diminishes is crucial for evaluating the reliability of multi-agent annotations. In this context, we also investigate whether increasing the number of debate

rounds helps reduce spurious agreements and thereby improves the reliability of DREAM's auto-labeling. The experiment in Appendix B.3 reveals that increasing the number of debate rounds does not consistently reduce the spurious consensus ratio.

### 3.2.3 ABLATION STUDY: DIVERSE DEBATE CONFIGURATIONS

It is of interest to examine how increased diversity in debate influences the auto-labeling performance of DREAM. Hence, we conduct an ablation study examining three factors: (i) increasing the number of agents, (ii) using heterogeneous LLM families, and (iii) adjusting temperature values. For the sake of space, we defer the details to AppendixB.4. To summarize, increasing diversity in the debating setup does not yield improved annotation performance. First, increasing the number of agents reduces the likelihood of reaching agreement on relevant cases, which lowers relevance recall and a slightly decreases overall annotation accuracy (See Appendix B.4.1). Second, although heterogeneous model families alleviate model-specific bias, the overall accuracy remains lower due to performance disparities across model types (See Appendix B.4.2). Third, higher temperature settings yield longer rationales, which cause the deliberation to drift toward irrelevant decision and ultimately degrade annotation quality (See Appendix B.4.3).

### 3.2.4 ABLATION STUDY: DEBATE HISTORY AS SUPPORTIVE RESOURCE

Table 3 shows the contribution of debate history to human annotation in uncertain escalated cases, contrasting annotation quality with and without it as a supportive resource. Each case is annotated by three MTurk crowdworkers with majority voting. The results demonstrate that debate history raises IAA, reflecting greater consistency even in difficult cases, and

Table 3: Comparison of human annotation quality with and without debate history.

| Method | IAA Fleiss $\kappa$ | Class-wise Recall | | bAcc |
|---|---|---|---|---|
| | | Irrelevance | Relevance | |
| wo. History | 0.50 | 84.9% | 89.6% | 87.3% |
| w. History | 0.62 | 89.3% | 94.6% | 92.0% |

therefore boosts labeling accuracy (bAcc). This highlights a unique advantage of DREAM over confidence-based methods, demonstrating genuine AI-human synergy.

## 4 REAL-WORLD APPLICATIONS TO IR BENCHMARK

To demonstrate the utility of our framework, we refine existing IR benchmarks using DREAM (Section 4.1). Then, we identify how holes in existing IR benchmarks introduce evaluation bias and demonstrate that our approach effectively mitigates this issue (Sections 4.2–4.3).

### 4.1 BRIDGE: IMPROVED IR BENCHMARK

A critical issue in IR benchmarks is the presence of holes (Craswell et al., 2020; Rassin et al., 2024), where truly relevant ("gold") chunks are missing from the labeled evaluation set and thus misclassified as irrelevant ("non-gold"). This mislabeling arises because the evaluation set is usually built from a candidate chunk pool—the union of top-ranked results of a few selected retrieval systems—primarily to reduce the human annotation cost (MacAvaney & Soldaini, 2023; Rassin et al., 2024). Such holes therefore cause an evaluation bias: the performance of retrieval models outside the pool are often underestimated, especially when the pool relies on limited or outdated retrieval systems.

To fill the labeling gap, we refine existing IR benchmarks with DREAM by adding missing relevance labels through large-scale yet low-cost labeling. *This process yields* 116,622 *relevance labels in total, of which* 29,824 *are identified as "relevant (holes)," at a human cost of only about* $506. This adds 428% more to the 6,976 originally annotated gold chunks, for a combined total of 36,800.

**Target Benchmark.** We refine two widely used IR benchmarking resources to address their inherent biases and improve evaluation reliability. In detail, we select seven IR benchmark subsets: MS MARCO (MS) and NQ from BEIR (Thakur et al., 2021); and Lifestyle (Life), Recreation (Rec), Science (Sci), Technology (Tech), and Writing (Writ) from ROBUSTQA (Han et al., 2023). These datasets span diverse domains and corpora of varying sizes and styles, providing a broad and challenging basis for retrieval evaluation. In making this selection, we also prioritize IR benchmarks with query–answer pairs, as this structure enables the construction of query–chunk–answer triplets, which in turn facilitates examining how improvements in retrieval evaluation translate to more reliable assessment of generation in RAG. We sample a total of 3,657 queries from the test sets of the seven subsets, with 550 queries from each dataset except for Science, which contains 357 valid queries.

Table 4: Hole@10 ratios across 25 retrieval systems on the target benchmark prior to refinement with `DREAM`. Systems with a higher percentage of holes than average are marked in **bold**. A detailed breakdown of the results for each of the seven subsets is in Appendix G.

| Sparse | | Dense | | | | | | | | | |
|--------|--------|------------|-------|--------|-------|-------|-------|-------|--------|-------------|--------|
| BM25 | Splade | Contriever | DPR | DimRed | SBERT | BPR | Faiss | ANCE | ColBERT | Aggretriever | Arctic |
| 14.6% | **19.0%** | 6.5% | 9.5% | 11.5% | 13.0% | 14.5% | 15.9% | 16.2% | 16.6% | **17.5%** | **20.8%** |

| BM25 + Rerank | | | | | ANCE + Rerank | | | | | BM25 + Rewrite | | |
|-------|--------|-------|-------|-------|-------|--------|-------|-------|-------|-------|-------|-------|
| TBERT | NBoost | MLM | ELEC | MT5 | TBERT | NBoost | MLM | ELEC | MT5 | HyDE | Q2d | MuGI |
| 17.3% | **18.5%** | **18.5%** | **18.6%** | **20.0%** | 17.7% | **18.8%** | **18.8%** | **18.8%** | **20.2%** | **21.2%** | **21.3%** | **22.3%** |

Abbreviation: DimRed=Dim Reduction, ColBERT=TCT-ColBERT, TBERT=TinyBERT, MLM=MiniLM, ELEC=ELECTRA, MT5=MonoT5, Q2d=Query2doc

The collected queries, together with the augmented relevance labels using `DREAM` (described in detail below), constitute our refined benchmark, which we name `BRIDGE` (see Appendix C).

**Candidate Pool Construction.** To fill the holes in the selected subsets, we construct a pool of 25 retrieval systems, combining both traditional retrievers and more recent ones introduced after the release of the target benchmarks, including sparse (*e.g.*, SPLADE (Formal et al., 2022)), dense (*e.g.*, Arctic (Yu et al., 2024)), and their combinations with advanced fusion techniques such as re-ranking (*e.g.*, Cross-Encoder (Li et al., 2024)) and query rewriting (*e.g.*, MuGI (Zhang et al., 2024a)), as summarized in Appendix D. Then, we combine the top-10 retrieved chunks from each of the 25 retrieval systems for selected 3,657 queries, followed by applying a simple LLM-based filter to remove typical irrelevant cases (see Appendix E). This yields 116,622 query–answer–chunk triplets (out of 296,053 initially retrieved), which constitute the candidate set for subsequent assessment.

**Relevance Assessment.** Each triplet in the candidate pool is considered as input to `DREAM`, under the same implementation described in Section 3.2. Through this process, 112,566 out of 116,622 cases reached consensus within two rounds of debate and were therefore automatically labeled with high certainty, while only the remaining 4,056 cases of disagreement required human evaluation. Hence, we perform human annotation using MTurk. Each disagreement case is evaluated by three qualified annotators, and the final label is decided based on the majority vote of their judgments. In our human annotation, the inter-annotator agreement (IAA) measured by Fleiss' Kappa is 0.62, indicating substantial agreement. The recruitment and procedures are detailed in Appendix F.

## 4.2 HOLES AS THE SOURCE OF EVALUATION BIAS

Our benchmark refinement with `DREAM` reveals that the original benchmarks contain a substantial number of holes with respect to the 25 retrieval systems we considered, indicating that many truly relevant chunks were left unlabeled. Table 4 presents the Hole@10 ratio, defined as the fraction of detected holes—relevant chunks newly identified—within the top-10 chunks retrieved by each system. The results show that the average Hole@10 across the 25 systems is 17.1%, which is considerably large. Among them, retrieval systems based on advanced fusion approaches, such as rerank and rewrite, as well as recent systems, such as Aggretriever and Arctic, exhibit higher Hole@10 ratios than other conventional ones. These highlight that current benchmarks largely underestimate the effectiveness of advanced retrievals due to pervasive unlabeled relevant chunks, leading to evaluation bias. Therefore, benchmark refinement is essential to ensure fair and reliable evaluation.

## 4.3 BIAS REDUCTION THROUGH FILLING THE HOLES

We validate the bias reduction of our benchmark refinement with `DREAM` through two metrics: (i) *growth rate*, which measures the proportion of newly detected holes as the number of retrievers in the candidate pool increases, and (ii) *marginal contribution*, which quantifies the incremental retrieval performance gain when a target retriever is added during this progressive expansion (see the equation in Appendix H). For the refinement to be effective, both metrics should converge toward zero, since no additional retrievers should uncover new holes and further additions should not affect retrieval performance. Figure 3 shows benchmark refinement through DREAM's hole-filling: (a) the growth rate of newly identified gold chunks quickly saturates as more retrievers are added, and (b) the averaged marginal contribution across all retrievers diminishes with increasing retriever diversity, converging near zero. It confirms our refinement eliminates major holes in the original benchmarks, mitigates bias, and ensures reliable evaluation for future retrievers.

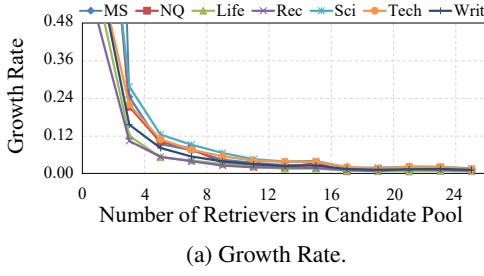
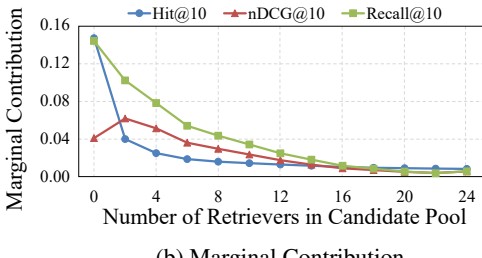

(a) Growth Rate.                                    (b) Marginal Contribution.

Figure 3: Bias reduction via incremental addition of retrievals in DREAM's hole filling: (a) reducing growth rate of newly identified holes across seven benchmark subsets as more systems are incorporated into the candidate pool; and (b) diminishing marginal contribution of a target retriever, measured by Hit@10, nDCG@10, and Recall@10, as the pool expands. To ensure statistical reliability, results are averaged over 10 runs, each with 25 retrieval systems added in random order.

Table 5: Retrieval performance using Hit@10 on BRIDGE for 25 retriever systems. Subscript numbers indicate improvements over the original benchmarks through hole filling. A detailed breakdown for each benchmark and results with nDCG@10 metric are provided in Tables 19 and 20.

| Sparse | | Dense | | | | | | | | | |
|---|---|---|---|---|---|---|---|---|---|---|---|
| BM25 | Splade | Contriever | DPR | DimRed | SBERT | BPR | Faiss | ANCE | ColBERT | Aggretriever | Arctic |
| 0.65 0.23 | 0.80 0.11 | 0.50 0.19 | 0.54 0.19 | 0.60 0.18 | 0.66 0.18 | 0.69 0.18 | 0.73 0.16 | 0.74 0.17 | 0.77 0.12 | 0.79 0.13 | 0.87 0.11 |

| BM25 + Rerank | | | | | ANCE + Rerank | | | | | BM25 + Rewrite | | |
|---|---|---|---|---|---|---|---|---|---|---|---|---|
| TBERT | NBoost | MLM | ELEC | MT5 | TBERT | NBoost | MLM | ELEC | MT5 | HyDE | Q2d | MuGI |
| 0.72 0.17 | 0.76 0.12 | 0.75 0.16 | 0.75 0.18 | 0.78 0.14 | 0.77 0.16 | 0.80 0.10 | 0.80 0.15 | 0.80 0.16 | 0.83 0.12 | 0.67 0.27 | 0.71 0.25 | 0.75 0.24 |

# 5  BENCHMARKING RETRIEVAL AND GENERATION WITH BRIDGE

We re-benchmark 25 retrieval systems on BRIDGE and extend the evaluation to RAG generation to examine how our hole filling impacts both retrieval and downstream generation performance.

## 5.1  RE-BENCHMARKING RETRIEVAL SYSTEMS AFTER HOLE FILLING

Table 5 presents the Hit@10 retrieval success rates of 25 systems on BRIDGE, with their score improvements over the original benchmark before hole filling with DREAM. All systems show higher Hit@10 scores after hole filling because previously missing relevant chunks are correctly recognized as gold. However, the improvement magnitude varies substantially across systems, which in turn introduces bias in IR evaluation when such holes remain unaddressed.

Among the retrieval systems, advanced fusion with re-writing (BM25 + Rewrite), which previously exhibited the highest Hole@10 ratios in Table 4, shows the largest improvement in Hit@10 scores after hole filling, indicating that systems with many missing relevant chunks are disproportionately penalized. This confirms that although prior benchmarks reveal evaluation bias due to holes, DREAM successfully fills these gaps with a 25-retriever pool. As shown in Figure 3, the growth rate of newly identified holes and the marginal contribution of additional retrievers converge to near zero, demonstrating that the bias is effectively resolved and enables fair evaluation over all retrieval systems.

We further analyze retrieval performance changes across the seven subsets in BRIDGE, covering diverse domains. The detailed results and analysis are presented in Appendix I.

## 5.2  ENHANCING RETRIEVAL-GENERATION ALIGNMENT IN RAG EVALUATION

Reliable assessment of RAG systems requires alignment between retrieval and generation. When IR performance is underestimated due to benchmark holes, the alignment between retrieval and generation breaks down, leading to biased system evaluation. For example, strong generation may be wrongly attributed to parametric knowledge because relevant chunks were mislabeled as non-relevant, causing retrieval to be counted as a failure even when it actually succeeded.

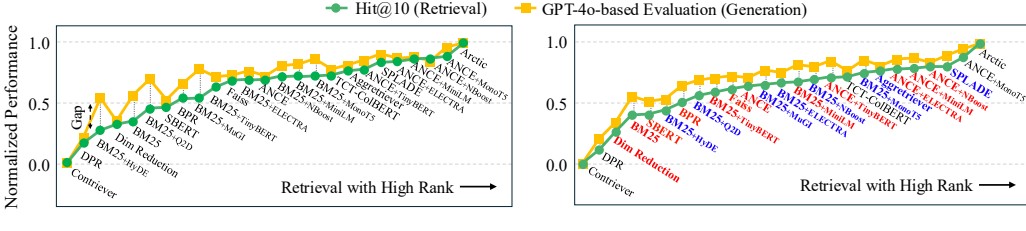

(a) Original Benchmark.       (b) BRIDGE Benchmark.

Figure 4: Normalized retrieval (green) and generation (yellow) performance across RAG systems, using the same Llama3.1-8B-Instruct model but different retrievers (labeled by name). Systems are sorted in ascending order of Hit@10 scores along the x-axis. Colors indicate rank changes between the original benchmark and BRIDGE: Red for drops and Blue for improvements, from (a) to (b).

To quantify this alignment, we introduce a metric, RAGAlign, which captures how improvements in retrieval translate into generation performance. It is defined for any pair of binary retrieval metric, METRICR@$K$ (*e.g.*, Hit@K), and binary generation metric, METRICG (*e.g.*, LLM-based binary evaluation). For each query, both metrics provide a binary outcome indicating success or failure for their respective tasks. Given $N$ queries in the benchmark data, the score is computed as: RAGAlign@K $= \frac{1}{N} \sum_{i=1}^{N} \mathbf{1} \left[ \text{METRICR}_i = \text{METRICG}_i \right]$, which is the proportion of queries where retrieval and generation outcomes agree, reflecting their alignment. We instantiate the metric with Hit@10 and GPT-4o–based binary evaluation as the two metrics (see Appendix J.3).

**Alignment Improvement in BRIDGE.** Table 6 reports RAGAlign@10 scores on BRIDGE, comparing results before and after hole filling with DREAM. For each query, we use the top-10 results from 25 retrieval systems, and the generation metric is computed from answers generated by Llama3.3-8B-Instruct for each query–chunk pair under a RAG pipeline (refer to Appendix J for details).The results show that hole filling

Table 6: RAGAlign@10 score gains over seven subsets of BRIDGE.

| Version | BEIR | | ROBUSTQA | | | | | Avg. |
|---|---|---|---|---|---|---|---|---|
| | MS | NQ | Life | Rec | Sci | Tech | Writ | |
| Original | 0.59 | 0.73 | 0.76 | 0.74 | 0.64 | 0.67 | 0.74 | 0.70 |
| BRIDGE | 0.88 | 0.86 | 0.84 | 0.81 | 0.85 | 0.84 | 0.81 | 0.84 |

with DREAM substantially strengthens retrieval–generation alignment in RAG evaluation. On average, RAGAlign@10 improves by 0.14 compared to the benchmark before refinement. That is, our benchmark, BRIDGE, provides a clearer link between retrieval and generation, addressing the limitation of earlier benchmarks. This trend remains consistent even when using other LLMs such as Qwen2.5-7B-Instruct and Gemma-2-9b-it. See Appendix J.3 for details.

**Impact of Better Alignment on RAG Evaluation.** Figure 4 compares normalized retrieval (green dots) and generation (yellow squares) performance across RAG systems, with (a) showing the original benchmarks and (b) the refined BRIDGE. After hole filling, substantial system-level rank shifts are observed: 20 out of 25 systems change their retrieval rankings, with some (*e.g.*, SBERT, ANCE) dropping in position and others (*e.g.*, BM25+MuGI, SPLADE) rising, indicating that the original benchmarks distorted retrieval system comparisons.

Beyond retrieval ranking shifts, the figure also suggests improved retrieval-generation alignment from (a) to (b). In the original benchmark (a), the green and yellow curves show noticeable gaps for several systems, while in BRIDGE (b), the two curves align more closely with smaller gaps. The strong alignment is also supported by a Pearson correlation coefficient of 0.985 in BRIDGE. This indicates that retrieval gains more reliably translate into generation performance and thus DREAM offers a stronger basis for the evaluation of RAG systems.

## 6 CONCLUSION

In this paper, we proposed DREAM, a debate-based relevance assessment pipeline that achieves high accuracy and effective AI-to-human escalation with minimal human intervention, without requiring additional training. Building on this pipeline, we constructed BRIDGE benchmark dataset by filling missing relevant chunk in existing datasets, thereby mitigating bias in IR evaluation. Through re-benchmarking 25 IR systems, we further demonstrate that unaddressed holes cause systematic retrieval underestimation and misalignment between retrieval and generation performance.

ETHICS STATEMENT

This paper addresses relevance labeling using LLM agents and raises no ethical concerns from a methodological standpoint. In conducting human annotation, we maintained clear communication with all crowd-sourced annotators. These annotators were compensated above-minimum wage compensation and provided explicit consent for their contributions to be used in our study. No personally identifiable information was collected throughout the process, ensuring complete annotator privacy. For details on human annotation are provided in Appendix F. We will also publicly release our `BRIDGE` benchmark dataset.

REPRODUCIBILITY STATEMENT

Every experiment was conducted on public datasets specified in Appendix C and the preprocessing procedure is described in Section 4.1. We also utilized publicly available retrieval systems as listed in Appendix D. The multi-round dabate-based labeling process and the answer generation for RAG evaluation were implemented using various LLMs, covering both open-source and proprietary categories. For open-source models, we utilized Llama3.1-series, Llama3.3-series and Qwen2.5-series. For proprietary models, we used GPT-4o-mini and GPT-4o through their paid API services. Detailed description about model configurations and checkpoints is provided in Appendix K. We will also publicly release our implementation of `DREAM`.

ACKNOWLEDGMENTS

This work was supported by the National Research Foundation of Korea (NRF) grant funded by Ministry of Science and ICT (No. RS-2022-NR068758 and No. RS-2024-00334343) and by the Institute of Information & communications Technology Planning & Evaluation (IITP) grant funded by Ministry of Science and ICT (No. RS-2024-00445087).

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

# A   AUTOMATED RELEVANCE LABELING DETAILS

## A.1   MULTI-AGENT MULTI-ROUND DEBATE DETAILS

The prompt templates used for our multi-round debate are provided in Tables 27- 28. Each template includes placeholders that are instantiated differently for the two agents. We designate each agent as "Agent A" or "Agent B" and use clear self-referential markers such as "You are Agent A" and "Agent A (You)" when the agent refers to itself, while using "Agent B" when referring to the counterpart. This explicit labeling maintains clear role separation and ensures that each agent consistently argues from its designated position while directly addressing and challenging the opposing viewpoint.

**Multi-Round Protocol.**   Table 27 provides the prompts for the first round debate. In the first round, each agent is assigned an initial stance and is made aware of it through the role-conditioning. Table 28 shows the prompts for the subsequent round debate. In subsequent rounds, before deciding a new reasoning and label, each agent must consult the debate history, including the previous round's reasoning outputs and the recorded label for that round, and then update, defend or rebut its position.

**Prompt Operation.**   For each debate turn, the agent (i) carefully reads the query and all candidate answers to ensure full comprehension; (ii) reads the given chunk; and (iii) determines whether the chunk fully supports any candidate answer under the evaluation guidelines. The guideline is intended to constrain the agent's role to an answer-aware query–chunk relevance assessment, ensuring that the agent only verifies whether a chunk fully supports any candidate answer to the target query. In particular, the rules explicitly prohibit judging answer correctness, query–answer validity, or answer–answer consistency, thereby narrowing the task to strict answer-aware relevance assessment. To ground the judgment, the agent must quote the decisive evidence from the chunk. Finally, the agents provide rationale covering their query-chunk interpretation, response to the opposing agent's position, and clear justification for the final decision.

## A.2   EXPERT ANNOTATION PROTOCOL

To evaluate relevance labeling methods in Section 3.2, we collect the ground-truth relevance labels (1: relevance, 0: irrelevance) for 700 query-chunk-answer triplets. We recruit seven graduate-level NLP specialists with complementary academic backgrounds (*i.e.*, computer science, artificial intelligence, business, economics, and social science) and compensate them $15 per hour. All annotators complete five practice examples to align on guideline interpretation. To facilitate adjudication, each triplet is paired with supporting and opposing reasons generated by Llama3.1-8B-Instruct, encouraging balanced consideration before final judgment. A senior researcher spot-checks a random subset to verify label consistency and resolve potential misjudgments in ambiguous cases. This protocol ensures high-quality ground-truth labels for evaluating pipeline accuracy. Through expert annotation, the 700 samples are identified as 212 relevant and 488 irrelevant chunks.

## A.3   EXPERIMENT IMPLEMENTATION DETAILS

For the `LLMJudge`, we adopt a single-agent setting using the prompt in Table 17. `DREAM` employs two agents prompted with Table 27- 28, engaging in up to two rounds of debate. If the two agents reach an agreement, their consensus is taken as the final label; otherwise, the case is escalated to a human annotator for adjudication. We implement `LARA` based on the official code released on GitHub[1], where we employ the simple binary prompt setting.

## A.4   LLM ADJUDICATION SETUP DETAILS

We detail our LLM-based adjudication setup in Section 3.2.2. Adjudication is triggered only when the two agents fail to reach consensus on a label. The adjudicator is implemented with Llama3.3-70B-Instruct with temperature 0.0. The full prompt is in Table 29. During adjudication, the adjudicator receives the debate outputs produced at the end of the corresponding round from both agents as discussion history, along with the query, answers, and candidate chunk.

---

[1]`https://github.com/RikiyaT/LARA`

Table 7: Detailed statistics of the seven datasets in BRIDGE. Each dataset is categorized by domain and chunk length, including the number of corpus and queries, average number of chunks and answers per query, and average word counts for queries, chunks, and answers.

| Dataset | Domain | Chunk Length | # Corpus | # Query | Avg. # C/Q | Avg. # A/Q | Avg. # Words | | |
|---------|--------|--------------|----------|---------|------------|------------|-------|-------|--------|
| | | | | | | | Query | Chunk | Answer |
| MS MARCO | Websearch | Short | 8,841,823 | 550 | 16.77 | 1.11 | 5.83 | 56.58 | 13.67 |
| NQ | | | 2,681,468 | 550 | 7.04 | 1.44 | 9.17 | 75.83 | 2.14 |
| Lifestyle | Lifestyle | Long | 119,461 | 550 | 6.61 | 1.47 | 10.26 | 152.77 | 9.46 |
| Recreation | Recreation | | 166,975 | 550 | 4.63 | 1.46 | 8.74 | 137.67 | 6.94 |
| Science | Science | | 1,000,000 | 357 | 16.09 | 1.37 | 8.82 | 127.74 | 7.95 |
| Technology | Technology | | 638,509 | 550 | 10.92 | 1.51 | 9.11 | 144.95 | 9.45 |
| Writing | Writing | | 199,994 | 550 | 8.52 | 1.57 | 8.26 | 120.45 | 8.34 |

Table 8: Dataset-level chunk statistics across the two-stage debate process. The table shows: (1) the dataset name, (2) the initial pool of candidate chunks to be labeled, (3) the number of chunks retained after irrelevant chunk filtering, and (4–7) the number of chunks that reached or failed to reach consensus after Rounds 1 and 2 of agent-based debate.

| Dataset | # Candidate Chunks | After Irrelevant Chunk Filtering | After Debate Round 1 | | After Debate Round 2 | |
|---------|-------------------|----------------------------------|----------------------|----|----------------------|----|
| | | | # Consensus | # Disagreement | # Consensus | # Disagreement |
| MS MARCO | 35,579 | 20,137 | 17,893 | 2,244 | 1,382 | 862 |
| NQ | 41,656 | 11,126 | 10,558 | 568 | 422 | 146 |
| Lifestyle | 42,043 | 14,553 | 13,159 | 1,394 | 979 | 415 |
| Recreation | 44,466 | 11,207 | 10,283 | 924 | 672 | 252 |
| Science | 33,966 | 19,011 | 16,525 | 2,486 | 1,646 | 840 |
| Technology | 51,104 | 20,876 | 18,491 | 2,384 | 1,552 | 832 |
| Writing | 47,239 | 19,712 | 17,529 | 2,182 | 1,473 | 709 |
| Total | 296,053 | 116,622 | 104,438 | 12,182 | 8,126 | 4,056 |

# B DETAILED STUDY ON DEBATING FRAMEWORK

## B.1 IMPACT OF STANCE ORDER

In Table 9, we compare the final debate labels for all consensus cases under the "Relevant-first (RF)" and "Irrelevant-first (IF)". Across 676 queries, 96.7% (654/676) of predictions remain identical after swapping the agent order, and only 3.3% (22/676) change. Importantly, the flips occur in both directions: eight cases switch from relevant to irrelevant, and 14 cases switch in the opposite direction. If the model were simply following the first agent, label changes would be heavily skewed toward one side. The nearly symmetric flip pattern instead demonstrates no detectable dependence on the initial stance or first-speaker order.

Table 9: Row corresponds to the final labels obtained when agents tart from the "Relevant-first (RF)" stance, and column corresponds to the final labels obtained when agents start from the "Irrelevant-first (IF)" stance. The value represents the how consistent the final labels are across the two settings.

| RF \ IF | Irrelevant | Relevant | Total |
|---------|-----------|----------|-------|
| Irrelevant | 393 | 14 | 407 |
| Relevant | 8 | 261 | 269 |
| Total | 401 | 275 | 676 |

Moreover, as shown in Table 10, the bAcc essentially stable across different reasoning orders, indicating that the overall evaluation quality is not influenced by which agent speaks first. This indicates that the debate outcomes are not affected by fist-speaker bias or order-dependent effects.

Table 10: Comparison of bAcc and escalation ratio between "Relevant-first (RF)" and "Irrelevant-first (IF)".

| Method | Recall (Relevance) | Recall (Irrelevance) | bAcc | Escalation Ratio |
|--------|-------------------|---------------------|------|------------------|
| Relevant First (RF) | 98.4% | 91.9% | 95.2% | 3.8% |
| Irrelevant First (IF) | 97.3% | 90.1% | 93.7% | 4.4% |

## B.2 LATENCY COMPARISON

To examine the trade-off between annotation accuracy and annotation cost, we compute the cost and latency per sample (i.e., a single query–chunk–answer triplet) and compare our method DREAM

against both `Human-Only` and `LLMJudge` in Table 11. We omit the cost report for `LARA`, as it relies on the same single-agent model as `LLMJudge` and therefore incurs identical cost and latency.

The human or API cost reports the human labor cost for `Human-Only` and API cost (based on input and output token costs under Deep Infra pricing) for `LLMJudge` and `DREAM`. Latency is measured in two settings: Non-parallel latency corresponds to a single API session, where we sum the generation times of all participating agents. In contrast, parallel latency corresponds to multiple API sessions, measured as the wall-clock time when all agents generate their outputs concurrently. The reported results are averaged over 147 samples and we used Llama-3.3-70B-Instruct for `LLMJudge` and `DREAM`.

Table 11: Cost and latency of labeling methods. We report human labor cost for `Human-Only` and API cost for both `LLMJudge` and `DREAM`. Non-parallel latency denotes processing time under a single API session, while parallel latency reflects execution with multiple concurrent API sessions.

| Method | Human or API Cost ($) | Non-parallel Latency (sec) | Parallel Latency (sec) | bAcc (%) | Escalation Ratio (%) |
|---|---|---|---|---|---|
| Human-Only | 0.1 | 30 | 30 | 93.8 | 100.0 |
| LLMJudge | 0.000066 | 1.292 | 1.292 | 73.9 | 0.0 |
| DREAM | 0.000470 | 8.583 | 4.291 | 95.2 | 3.5 |

`Human-Only` yields strong labeling accuracy (bAcc), but this performance comes with very high human cost and latency. On the other hand, `LLMJudge` minimizes the cost and latency but its bAcc is significantly lower than `Human-Only`, indicating that it struggles to reliably identify relevance labels. In contrast, `DREAM` achieves even higher bAcc while being 200x cheaper and 3.5x–7.0x faster than `Human-Only`. This shows that `DREAM` achieves a favorable balance between cost and performance, enabling scalable high-quality annotation that would be infeasible with humans alone.

## B.3 Agreement Stability and Annotation Quality across Multi-round Debate

A central validity concern in multi-agent debates is whether consensus reflects genuine convergence or spurious agreement due to stochastic fluctuations. To assess the stability of consensus, we examine whether round-1 agreements persist when the debate continues for additional rounds, and we compute the corresponding spurious consensus rate. We sample 651 instances that reach consensus in round 1 and continue the debate up to round 5, with discussion history consisting only of previous-round outputs.

Table 12: Retention of round 1 consensus and spurious consensus rate across subsequent rounds (R2–R5). Persistence ratio denotes the proportion of cases that remain in consensus in each later round.

| Round | R1 | R2 | R3 | R4 | R5 |
|---|---|---|---|---|---|
| Persistence ratio (%) | - | 96.5 | 96.2 | 96.2 | 96.0 |
| # Total cases | 651 | 651 | 651 | 651 | 651 |
| # Consensus | 651 | 628 | 626 | 626 | 625 |
| Spurious consensus rate (%) | 5.0% | 3.9% | 5.0 % | 5.1% | 5.1% |

Table 12 shows that 96% of round 1 consensus is preserved in round 2, and this level of stability continues through round 5, indicating that the consensus can be regarded as stable. However, increasing the number of debate rounds does not consistently reduce the spurious consensus rate. The lowest rate appears at R2 (3.9%), after which it rises again, reaching 5.1% at R5. That is, more rounds eventually introduce additional noise into the discussion, making incorrect agreements more likely rather than less.

In detail, from R1 to R2, we observe cases where spurious consensuses or initial disagreements are corrected in R2, because these errors are relatively easy for LLMs to reassess once the opposing arguments are explicitly surfaced. However, extending the debate beyond two rounds pushes the cases that remain ambiguous even after R2 toward an incorrect "irrelevance" consensus, as evidenced by the drop in recall (relevance) while irrelevance recall remains stable. These results demonstrate that using more than two rounds carries a risk of amplifying spurious consensus.

## B.4 Impact of Improved Diversity

`DREAM` uses two Llama3.3-70B-Instruct models with a temperature of 0.0 as its LLM agent. To investigate whether increasing diversity in the debating setup improves annotation quality, we conduct an ablation study with `DREAM` on its two design choices: (i) increasing the number of agents, (ii) using heterogeneous model families, and (iii) using non-zero temperature for each agent.

### B.4.1 INCREASING AGENT NUMBER

We first examine the impact of scaling the number of debating agents from two to four. Specifically, we compare three configurations: (1) `2-Llama`, which uses two Llama3.3-70B-Instruct models (our default setting); (2) `4-Llama`, which employs four Llama3.3-70B-Instruct models; and (3) `2-Llama + 2-Qwen`, which combines two Llama3.3-70B-Instruct and two Qwen2.5-72B-Instruct models, with each model family initialized with both relevant and irrelevant stances.

Table 13: Comparison of annotation quality under different numbers of debate agents. `2L+2Q` represents a heterogeneous configuration combining two Llama and two Qwen models.

| Method | Class-wise Recall | | bAcc | Escalation Ratio |
|---|---|---|---|---|
| | Irrelevance | Relevance | | |
| `2-Llama` | 91.9% | 98.4% | 95.2% | 3.5% |
| `4-Llama` | 97.2% | 91.4% | 94.3% | 11.2% |
| `2L + 2Q` | 97.3% | 92.0% | 94.7% | 13.0% |

As expected, increasing the number of agents naturally reduces the consensus rate ($100\% -$ escalation ratio), since more models must reach agreement before a final label can be assigned. Although this stronger agreement requirement could be expected to improve labeling accuracy (bAcc) on non-escalated cases, we find that bAcc actually decreases. As more models participate, the debating process becomes increasingly conservative toward relevant cases, making agreement on them harder to reach compared to irrelevant ones. This leads to a higher number of disagreements on relevant examples, which in turn reduces recall for the relevant class and overall labeling accuracy.

### B.4.2 USING HETEROGENEOUS LLMS

We next investigate whether introducing heterogeneity in model families can mitigate inherent biases and improve annotation quality. We evaluate four configurations: (1) `2-Llama` (our default); (2) `2-Qwen`, using two Qwen2.5-72B-Instruct models; (3) `Llama(Relevant) + Qwen(Irrelevant)`, where Llama is initialized with the relevant stance and Qwen with the irrelevant stance; and (4) `Qwen(Relevant) + Llama(Irrelevant)`, the reverse configuration of (3).

Table 14: Comparison of annotation quality across debate settings with different model families. `L+Q` uses Llama as the relevant initiator and Qwen as the irrelevant initiator; `Q+L` uses the opposite initialization.

| Method | Class-wise Recall | | bAcc | Escalation Ratio |
|---|---|---|---|---|
| | Irrelevance | Relevance | | |
| `2-Llama` | 91.9% | 98.4% | 95.2% | 3.5% |
| `2-Qwen` | 95.1% | 82.8% | 89.0% | 2.6% |
| `L + Q` | 95.8% | 88.7% | 92.3% | 6.6% |
| `Q + L` | 96.0% | 91.4% | 93.7% | 6.5% |

Our analysis reveals distinct biases in each model family: the `2-Llama` baseline exhibits higher recall for the relevant class, while the `2-Qwen` baseline shows higher recall for the irrelevant class, indicating that Llama leans toward predicting "relevant" and Qwen toward "irrelevant". The heterogeneous setup reduces these biases, achieving higher recall for the irrelevant class than `2-Llama` and higher recall for the relevance class than `2-Qwen`. However, its bAcc remains lower than `2-Llama` because Qwen's weaker performance drags down overall accuracy. This suggests that heterogeneous debate is beneficial only when all participating models are individually strong.

### B.4.3 ADJUSTING TEMPERATURE VALUES

We also vary the sampling temperature under the 2-Llama configuration to assess its impact on debate quality. Under higher temperatures, models produce often longer reasoning because the sampling distribution becomes flatter, making low-probability tokens more likely to be selected. This lengthening causes the debate to drift toward irrelevance, which in turn makes the final decision more conservative. Although recall for the irrelevant class improves slightly, the substantial drop in recall for the relevant class leads to an overall decline in labeling accuracy (bAcc).

Table 15: Comparison of annotation quality across debate settings with different model temperatures.

| Temp. | Class-wise Recall | | bAcc | Escalation Ratio |
|---|---|---|---|---|
| | Irrelevance | Relevance | | |
| `0.0` | 91.9% | 98.4% | 95.2% | 3.5% |
| `0.6` | 93.5% | 92.7% | 93.1% | 3.8% |
| `1.0` | 93.3% | 91.9% | 92.6% | 3.1% |

Table 16: Publicly available retrieval model links used in this paper.

| Model | Public Model Checkpoint (Link) |
|---|---|
| BM25 (Robertson et al., 2009) | https://pypi.org/project/pyserini/ |
| SBERT (Reimers & Gurevych, 2019) | https://huggingface.co/sentence-transformers/msmarco-distilbert-base-tas-b |
| NBoost (Thienes & Pertschuk, 2019) | https://huggingface.co/nboost/pt-bert-base-uncased-msmarco |
| DPR (Karpukhin et al., 2020) | https://huggingface.co/sentence-transformers/facebook-dpr-ctx-encoder-multiset-base |
| Faiss Dense (Karpukhin et al., 2020) | https://github.com/facebookresearch/faiss/wiki |
| TinyBERT (Jiao et al., 2020) | https://huggingface.co/cross-encoder/ms-marco-TinyBERT-L-6 |
| MiniLM (Wang et al., 2020) | https://huggingface.co/cross-encoder/ms-marco-MiniLM-L12-v2 |
| MonoT5 (Nogueira et al., 2020) | https://huggingface.co/castorini/monot5-base-msmarco |
| BPR (Yamada et al., 2021) | https://huggingface.co/income/bpr-gpl-trec-covid-base-msmarco-distilbert-tas-b |
| Contriever (Izacard et al., 2021) | https://huggingface.co/facebook/contriever |
| TCT-COLBERT (Lin et al., 2021) | https://huggingface.co/castorini/tctcolbert-v2-hnp-msmarco |
| ANCE (Xiong et al., 2021) | https://huggingface.co/sentence-transformers/msmarco-roberta-base-ance-firstp |
| Dim Reduction (Liu et al., 2022) | https://huggingface.co/sentence-transformers/msmarco-distilbert-base-tas-b |
| SPLADE (Formal et al., 2022) | https://huggingface.co/naver/splade-cocondenser-ensembledistil |
| Aggretriever (Lin et al., 2022) | https://huggingface.co/castorini/aggretriever-cocondenser |
| ELECTRA (Li et al., 2023) | https://huggingface.co/cross-encoder/ms-marco-electra-base |
| HyDE (Gao et al., 2023a) | https://github.com/texttron/hyde |
| Q2D (Wang et al., 2023) | https://arxiv.org/pdf/2303.07678 |
| MuGI (Zhang et al., 2024b) | https://github.com/lezhang7/Retrieval-MuGI |
| Arctic (Merrick et al., 2024) | https://huggingface.co/Snowflake/snowflake-arctic-embed-m |

## C  DETAIL OF BRIDGE BENCHMARK DATASET

### C.1  SOURCE DATA

MS MARCO (Nguyen et al., 2016) contains real anonymized Bing queries paired with manually curated answers reflecting real-world search behavior. NQ (Kwiatkowski et al., 2019) consists of Google queries linked to Wikipedia pages with human-annotated answer spans. LoTTE (Santhanam et al., 2022) provides long-tail, topic-stratified QA data from the StackExchange community, which enables robust out-of-domain evaluation. It covers five domains: Lifestyle (*e.g.*, cooking, sports, and travel), Recreation (*e.g.*, gaming, anime, and movies), Science (*e.g.*, math, physics, and biology), Technology (*e.g.*, Apple, Android, UNIX, and security), and Writing (*e.g.*, English). These domains introduce diverse question styles and content types, offering a comprehensive evaluation setting that spans both factual and user-generated queries.

### C.2  BRIDGE STATISTICS DETAIL

Table 7 presents detailed statistics of the 7 datasets in BRIDGE, spanning 6 diverse domains including web search, science, and lifestyle. The datasets are categorized as either short- or long-form based on the average chunk length, with those exceeding 100 words per chunk classified as long-form. The benchmark contains 3,657 queries. Each domain contributes 550 queries, and each query is paired with an average of 10.08 gold chunks. Answer lengths range from 2.14 to 13.67 words, covering both concise and elaborate responses. We also provide the chunk statistics observed during the relevant assessment, spanning from candidate chunk pooling to debate round 2, in Table 8.

## D  RETRIEVAL SYSTEMS

Throughout BRIDGE's construction and experimentation, we utilize 25 RAG systems, as described in Section 4.1. This configuration covers sparse, dense retrievers, as well as fusion techniques such as re-ranking and query rewriting. This varied setup facilitates comprehensive analysis in our experimental evaluations. These are publicly available retrieval models and methods (see Table 16).

## E  IRRELEVANT CHUNK FILTERING

To enhance the efficiency by filtering out obviously irrelevant cases, we discard clearly irrelevant chunks prior to conducting relevance assessment. It employs three LLMs, *i.e.*, GPT-4o-mini, Llama3.3-70B-Instruct, and Qwen2.5-72B-Instruct, as it helps reduce the risk of over-pruning due to single-model bias (Owens et al., 2024; Borah & Mihalcea, 2024). The prompt is presented in Table 17. Since the dataset consists of query-chunk-answer triplets, we consider both query and

Table 17: Prompt for `LLMJudge` and irrelevant chunk filtering.

---

Determine if the DOCUMENT contains enough information to answer the QUERY with any of the AN-SWERs.

QUERY: {query}
DOCUMENT: {chunk}
ANSWERs:
{ground-truth answers}

If any ANSWER is supported by the DOCUMENT, return `true`. If all ANSWERs are not supported, return `false`. Do not provide explanations or additional information.

Respond strictly in this format:
`{"is_supported":  true/false}`

---

ground-truth answer when assessing chunk relevance. Each chunk is evaluated by three LLMs for sufficiency in answering the query given the ground-truth answer, and discarded only if all agree it is irrelevant, thereby reducing the candidate pool from 296,053 to 116,622.

## F  HUMAN ANNOTATION

### F.1  RECRUITMENT AND COMPENSATION

We apply strict qualification criteria for MTurk annotators to ensure the quality of human annotation. Participation is restricted to MTurk workers with a HIT approval rate above 90% and at least 500 approved HITs to filter for experienced workers. Additionally, all annotators must achieve a minimum score of 90 on our custom English comprehension exam designed to simulate actual annotation tasks, ensuring both fluency in English and familiarity with the task format.

Annotators are compensated $7.5 per hour, a rate that exceeds the U.S. federal minimum wage. All procedures follow standard ethical guidelines, including informed consent, with instructions clearly stating that annotations will be used solely for academic research and consent obtained prior to participation. To protect worker privacy, we do not collect any personal or sensitive data, including detailed demographics such as age or gender. As the study poses minimal risk and ensures participant anonymity, we did not seek ethics board approval in accordance with standards commonly considered exempt from review.

### F.2  ANNOTATION PROCESS

**Human Annotation.** Each chunk is evaluated by three qualified MTurk workers, who are provided with the target query, a set of its possible answer(s), the candidate chunk, and the debating history that produced disagreement. Annotators are instructed to determine whether the chunk contains sufficient evidence to support the gold answer(s), following a unified annotation guideline. The debating history is provided to help annotators understand why the agents led to disagreement and to highlight potential ambiguities, thereby enabling more informed and consistent labeling decisions. Final labels are determined by majority vote among the three annotators.

**Attention Check.**  We insert 10% of attention checks into each HIT. These are designed to resemble real tasks, but have objectively verifiable answers to ensure high-quality annotations on crowd-sourcing platforms. Specifically, we construct attention checks by pairing a ground-truth chunk from existing datasets with a query–gold answer pair and two conflicting arguments from unrelated domains. Attention check chunks are intentionally correct and must be labeled as relevant candidate chunks. Annotators are required to pass all attention checks for their responses to be accepted. Attention checks are randomly placed to prevent predictability and have moderate difficulty to maintain both fairness and effective filtering. This strict filtering improves overall label consistency without significantly increasing cost.

## G HOLES AS THE SOURCE OF EVALUATION

Table 18 shows a detailed breakdown of the Hole@10 ratios across 25 retrieval systems for each of the seven subsets. The results indicate that MS MARCO exhibits the highest Hole@10, which can be attributed to its large corpus size.

Table 18: Hole@10 ratios across 25 retrieval systems and the existing benchmark. Systems with a higher percentage of holes than the average across retrieval systems are marked in **bold**.

| Method | Model | | MS | NQ | Life | Rec | Sci | Tech | Writ | Avg. |
|--------|-------|---|----|----|------|-----|-----|------|------|------|
| Sparse | BM25 | | 23.3% | 12.3% | 10.5% | 7.1% | **21.5%** | **14.2%** | **13.3%** | 14.6% |
| | SPLADE | | **41.3%** | **18.2%** | **12.4%** | **8.8%** | **23.0%** | **15.3%** | **14.3%** | **19.0%** |
| Dense | Contriever | | 16.1% | 5.1% | 3.9% | 3.7% | 6.4% | 4.9% | 5.7% | 6.5% |
| | DPR | | 22.6% | 12.4% | 5.2% | 3.2% | 10.6% | 5.2% | 7.5% | 9.5% |
| | Dim Reduction | | 35.0% | 9.0% | 8.1% | 5.0% | 9.4% | 7.3% | 6.7% | 11.5% |
| | SBERT | | 35.5% | 10.5% | 8.1% | 5.6% | 14.4% | 8.4% | 8.7% | 13.0% |
| | BPR | | 35.9% | 12.9% | 8.8% | 6.2% | 16.4% | 10.6% | 10.5% | 14.5% |
| | Faiss | | **39.5%** | 14.3% | 10.7% | 7.4% | 17.7% | 11.6% | 10.3% | 15.9% |
| | ANCE | | **37.8%** | 13.4% | 11.2% | 7.4% | 18.3% | 12.5% | 12.6% | 16.2% |
| | TCT-COLBERT | | **39.2%** | 15.4% | 10.8% | 7.1% | 19.1% | 12.7% | 12.2% | 16.6% |
| | Aggretriever | | **39.9%** | 15.7% | 10.8% | **8.3%** | 20.2% | **14.2%** | **13.4%** | 17.5% |
| | Arctic | | 41.1% | **18.4%** | **13.4%** | **10.9%** | 24.7% | 20.2% | 17.2% | 20.8% |
| Rerank | BM25 | TinyBERT | 35.8% | **16.2%** | **11.7%** | **8.2%** | **21.7%** | **14.1%** | **13.3%** | **17.3%** |
| | | NBoost | 37.2% | **17.8%** | **11.9%** | **9.7%** | **23.1%** | **14.9%** | **15.1%** | **18.5%** |
| | | MiniLM | 38.3% | **18.2%** | **12.0%** | **9.5%** | **22.3%** | **14.5%** | **14.6%** | **18.5%** |
| | | ELECTRA | 36.9% | **17.9%** | **13.0%** | **9.4%** | **23.4%** | **14.9%** | **14.4%** | **18.6%** |
| | | MonoT5 | 38.2% | **18.8%** | **13.3%** | **10.3%** | **25.6%** | **17.4%** | **16.1%** | **20.0%** |
| | ANCE | TinyBERT | **39.2%** | 15.9% | **12.0%** | **8.4%** | **21.1%** | **14.1%** | **13.3%** | **17.7%** |
| | | NBoost | **39.2%** | 17.4% | **12.7%** | **9.5%** | **22.8%** | **15.3%** | **14.7%** | **18.8%** |
| | | MiniLM | **41.3%** | **17.7%** | **12.7%** | **9.1%** | **21.4%** | **15.0%** | **14.5%** | **18.8%** |
| | | ELECTRA | **39.7%** | **17.7%** | **13.2%** | **9.2%** | **22.4%** | **15.1%** | **14.5%** | **18.8%** |
| | | MonoT5 | 40.4% | **18.2%** | **14.1%** | **10.2%** | **25.5%** | **17.3%** | **15.6%** | **20.2%** |
| Rewrite | BM25 | HyDE | 36.1% | **25.2%** | **14.0%** | **8.7%** | **28.0%** | **20.9%** | **15.7%** | **21.2%** |
| | | Q2D | 36.9% | **22.3%** | **14.4%** | **9.1%** | **27.5%** | **20.7%** | **18.0%** | **21.3%** |
| | | MuGI | 38.0% | **24.8%** | **14.8%** | **9.1%** | **28.7%** | **21.3%** | **19.6%** | **22.3%** |
| | Avg. | | 36.2% | 16.3% | 11.4% | 7.9% | 20.6% | 14.1% | 13.2% | 17.1% |

## H EVALUATION FORMULA

To demonstrate the bias reduction of `BRIDGE` refined by `DREAM`, we measured on two metrics.

**Growth Rate.** To assess whether the candidate pool constructed with multiple retriever is saturated enough, we compute the growth rate that is the proportion of newly detected relevance chunks (holes) as the number of retrievers in the candidate pool increases.

$$\text{GR}(m) = \frac{\left|\mathcal{H}(C_m)\right| - \left|\mathcal{H}(C_{m-1})\right|}{\left|\mathcal{H}(C_{m-1})\right|}, \qquad m \geq 1,$$

where $C_m$ denotes a candidate pool constructed with subset of $m$ retrievers, $\mathcal{H}(C_m)$ denotes the set of unlabeled relevant chunks discovered by the candidate pool $C_m$, $\mathcal{H}(C_{m-1})$ denotes the set of unlabeled relevant chunks discovered by the previous candidate pool $C_{m-1}$.

**Marginal Contribution.** To examine whether the dataset annotated with multiple retrievers remains robust for evaluating a new retriever not included in candidate pool, we evaluate each retriever on a dataset annotated with the results of the other 24 retrievers, excluding itself. The marginal contribution is defined as the absolute performance difference between the dataset annotated with the target retriever and the dataset where it is excluded. The marginal contributions across all retrievers are then averaged and illustrated in Figure 3.

$$\text{MC}_m(r) = \left| \text{Perf}_{D_r}(r) - \text{Perf}_{D_{S_m}}(r) \right|, \quad S_m \subseteq \mathcal{R} \setminus \{r\},$$

Table 19: Retrieval Hit@10 performance on `BRIDGE` for 25 retrievers over 7 datasets, with improvements (↑) over original benchmarks in subscripts. Best performance in **bold**.

| Method | Model | Dataset | | | | | | | Avg. |
|---|---|---|---|---|---|---|---|---|---|
| | | MS | NQ | Life | Rec | Sci | Tech | Writ | |
| Sparse | BM25 | 0.70 0.36 ↑ | 0.63 0.21 ↑ | 0.68 0.16 ↑ | 0.61 0.11 ↑ | 0.68 0.34 ↑ | 0.50 0.23 ↑ | 0.77 0.19 ↑ | 0.65 0.23 ↑ |
| | SPLADE | **0.89** 0.25 ↑ | 0.82 0.13 ↑ | 0.81 0.02 ↑ | 0.77 0.02 ↑ | 0.81 0.18 ↑ | 0.66 0.12 ↑ | 0.87 0.05 ↑ | 0.80 0.11 ↑ |
| Dense | Contriever | 0.61 0.41 ↑ | 0.48 0.21 ↑ | 0.52 0.08 ↑ | 0.51 0.06 ↑ | 0.45 0.23 ↑ | 0.38 0.26 ↑ | 0.58 0.08 ↑ | 0.50 0.19 ↑ |
| | DPR | 0.61 0.35 ↑ | 0.72 0.12 ↑ | 0.57 0.16 ↑ | 0.44 0.09 ↑ | 0.50 0.26 ↑ | 0.32 0.17 ↑ | 0.64 0.17 ↑ | 0.54 0.19 ↑ |
| | Dim Reduction | 0.79 0.32 ↑ | 0.56 0.15 ↑ | 0.70 0.13 ↑ | 0.59 0.10 ↑ | 0.52 0.26 ↑ | 0.45 0.18 ↑ | 0.58 0.14 ↑ | 0.60 0.18 ↑ |
| | SBERT | 0.84 0.30 ↑ | 0.66 0.15 ↑ | 0.66 0.10 ↑ | 0.63 0.11 ↑ | 0.61 0.24 ↑ | 0.49 0.18 ↑ | 0.70 0.18 ↑ | 0.66 0.18 ↑ |
| | BPR | 0.82 0.28 ↑ | 0.71 0.16 ↑ | 0.70 0.13 ↑ | 0.67 0.08 ↑ | 0.66 0.28 ↑ | 0.51 0.18 ↑ | 0.75 0.15 ↑ | 0.69 0.18 ↑ |
| | Faiss | 0.87 0.23 ↑ | 0.74 0.12 ↑ | 0.77 0.10 ↑ | 0.72 0.10 ↑ | 0.68 0.22 ↑ | 0.58 0.19 ↑ | 0.74 0.13 ↑ | 0.73 0.16 ↑ |
| | ANCE | 0.85 0.28 ↑ | 0.72 0.15 ↑ | 0.78 0.09 ↑ | 0.74 0.11 ↑ | 0.74 0.24 ↑ | 0.57 0.17 ↑ | 0.77 0.13 ↑ | 0.74 0.17 ↑ |
| | TCT-COLBERT | **0.89** 0.24 ↑ | 0.78 0.13 ↑ | 0.80 0.04 ↑ | 0.74 0.03 ↑ | 0.76 0.21 ↑ | 0.61 0.16 ↑ | 0.81 0.06 ↑ | 0.77 0.12 ↑ |
| | Aggretriever | 0.88 0.29 ↑ | 0.79 0.14 ↑ | 0.80 0.03 ↑ | 0.75 0.04 ↑ | 0.77 0.20 ↑ | 0.68 0.14 ↑ | 0.85 0.06 ↑ | 0.79 0.13 ↑ |
| | Arctic | **0.89** 0.32 ↑ | **0.86** 0.10 ↑ | **0.89** 0.02 ↑ | **0.89** 0.01 ↑ | **0.84** 0.15 ↑ | **0.81** 0.12 ↑ | **0.93** 0.04 ↑ | **0.87** 0.11 ↑ |
| Rerank / BM25 | TBERT | 0.80 0.28 ↑ | 0.72 0.12 ↑ | 0.74 0.13 ↑ | 0.71 0.08 ↑ | 0.72 0.25 ↑ | 0.55 0.20 ↑ | 0.80 0.15 ↑ | 0.72 0.17 ↑ |
| | NBoost | 0.82 0.30 ↑ | 0.75 0.12 ↑ | 0.79 0.02 ↑ | 0.76 0.03 ↑ | 0.77 0.21 ↑ | 0.61 0.13 ↑ | 0.84 0.05 ↑ | 0.76 0.12 ↑ |
| | MiniLM | 0.83 0.28 ↑ | 0.75 0.12 ↑ | 0.79 0.12 ↑ | 0.75 0.06 ↑ | 0.74 0.24 ↑ | 0.58 0.17 ↑ | 0.84 0.14 ↑ | 0.75 0.16 ↑ |
| | ELEC | 0.82 0.30 ↑ | 0.75 0.13 ↑ | 0.78 0.12 ↑ | 0.75 0.08 ↑ | 0.76 0.28 ↑ | 0.57 0.18 ↑ | 0.83 0.14 ↑ | 0.75 0.18 ↑ |
| | MonoT5 | 0.84 0.32 ↑ | 0.77 0.13 ↑ | 0.81 0.04 ↑ | 0.77 0.03 ↑ | 0.80 0.23 ↑ | 0.62 0.14 ↑ | 0.86 0.07 ↑ | 0.78 0.14 ↑ |
| Rerank / ANCE | TBERT | 0.86 0.26 ↑ | 0.77 0.12 ↑ | 0.79 0.11 ↑ | 0.75 0.08 ↑ | 0.76 0.23 ↑ | 0.60 0.19 ↑ | 0.83 0.13 ↑ | 0.77 0.16 ↑ |
| | NBoost | 0.85 0.26 ↑ | 0.81 0.13 ↑ | 0.85 0.01 ↑ | 0.79 0.00 ↑ | 0.79 0.15 ↑ | 0.66 0.11 ↑ | 0.86 0.02 ↑ | 0.80 0.10 ↑ |
| | MiniLM | 0.87 0.24 ↑ | 0.80 0.12 ↑ | 0.84 0.09 ↑ | 0.77 0.07 ↑ | 0.78 0.23 ↑ | 0.65 0.18 ↑ | 0.86 0.11 ↑ | 0.80 0.15 ↑ |
| | ELEC | 0.87 0.27 ↑ | 0.80 0.13 ↑ | 0.84 0.10 ↑ | 0.78 0.08 ↑ | 0.78 0.22 ↑ | 0.64 0.18 ↑ | 0.86 0.12 ↑ | 0.80 0.16 ↑ |
| | MonoT5 | **0.89** 0.31 ↑ | 0.82 0.12 ↑ | 0.87 0.02 ↑ | 0.81 0.02 ↑ | 0.83 0.17 ↑ | 0.70 0.13 ↑ | 0.88 0.04 ↑ | 0.83 0.12 ↑ |
| Rewrite / BM25 | HyDE | 0.75 0.42 ↑ | 0.79 0.16 ↑ | 0.66 0.21 ↑ | 0.58 0.15 ↑ | 0.67 0.39 ↑ | 0.53 0.31 ↑ | 0.69 0.23 ↑ | 0.67 0.27 ↑ |
| | Q2D | 0.78 0.40 ↑ | 0.81 0.15 ↑ | 0.72 0.19 ↑ | 0.63 0.12 ↑ | 0.68 0.37 ↑ | 0.58 0.31 ↑ | 0.79 0.20 ↑ | 0.71 0.25 ↑ |
| | MuGI | 0.81 0.39 ↑ | 0.83 0.13 ↑ | 0.75 0.18 ↑ | 0.68 0.12 ↑ | 0.71 0.36 ↑ | 0.61 0.31 ↑ | 0.83 0.19 ↑ | 0.75 0.24 ↑ |
| Avg. | | 0.82 0.31 ↑ | 0.75 0.14 ↑ | 0.76 0.10 ↑ | 0.70 0.07 ↑ | 0.71 0.25 ↑ | 0.58 0.19 ↑ | 0.79 0.12 ↑ | 0.73 0.17 ↑ |

where $\mathcal{R}$ denotes the full set of retrievers, $r \in \mathcal{R}$ is the target retriever, $S_m$ is a subset of $m$ retrievers excluding $r$, $D_A$ denotes a dataset annotated with retriever set $A$, and $\text{Perf}_D(r)$ denotes the performance of retriever $r$ on dataset $D$ (*e.g.*, Recall@k, Hit@k, nDCG@k).

## I   RETRIEVAL BENCHMARKING DETAIL

The comprehensive retrieval evaluation presented in Tables 19 and 20 demonstrates contrasting patterns between Hit@10 and nDCG@10 metrics following label correction. While Table 19 shows consistent positive improvements across all 25 retrieval systems, with query rewriting approaches (HyDE, Q2D, MuGI) achieving the most substantial gains averaging 0.25 to 0.27, Table 20 reveals a markedly different pattern. Most retrieval methods exhibit performance decreases in nDCG@10, with only MS MARCO showing consistent improvements and query rewriting methods demonstrating modest gains. This phenomenon can be attributed to the nDCG metric's structural characteristics. When additional relevant chunks are identified at lower ranking positions through label correction, the ideal DCG (IDCG) increases more substantially than the actual DCG, resulting in lower nDCG@k scores despite improved recall. These findings suggest that while the original benchmark systematically underestimated retrieval performance as evidenced by Hit@10 improvements, nDCG@10 may not be well-suited for evaluation scenarios with expanded ground truth annotations, particularly when multiple relevant chunks are present.

## J   ALIGNMENT EVALUATION DETAILS

This section provides detailed formulations of the metrics used to evaluate retrieval performance, generation quality, and their alignment in RAG systems. We employ binary retrieval metrics (Hit@k), non-binary retrieval metrics (nDCG@k), and GPT-4o-based binary evaluation for generation assessment.

Table 20: Retrieval nDCG@10 performance on `BRIDGE` for 25 retrievers evaluated on 7 datasets, with improvements (↑) over original benchmarks in subscripts. Best performance in **bold**.

| Method | Model | | Dataset | | | | | | | Avg. |
|--------|-------|------|------|------|------|------|------|------|------|------|
| | | | MS | NQ | Life | Rec | Sci | Tech | Writ | |
| Sparse | BM25 | | 0.25 0.07 ↑ | 0.27 0.02 ↑ | 0.32 -0.04 ↓ | 0.30 -0.03 ↓ | 0.22 0.01 ↑ | 0.17 0.02 ↑ | 0.35 -0.05 ↓ | 0.27 0.00 ↑ |
| | SPLADE | | **0.46** 0.08 ↑ | 0.45 -0.03 ↓ | 0.43 -0.10 ↓ | 0.43 -0.08 ↓ | 0.27 -0.10 ↓ | 0.24 -0.04 ↓ | 0.42 -0.14 ↓ | 0.39 -0.06 ↓ |
| Dense | Contriever | | 0.16 0.05 ↑ | 0.15 0.00 ↑ | 0.19 -0.04 ↓ | 0.21 -0.06 ↓ | 0.08 -0.03 ↓ | 0.08 -0.01 ↓ | 0.19 -0.06 ↓ | 0.15 -0.02 ↓ |
| | DPR | | 0.20 0.06 ↑ | 0.35 -0.07 ↓ | 0.22 -0.05 ↓ | 0.18 -0.04 ↓ | 0.11 -0.03 ↓ | 0.08 -0.01 ↓ | 0.23 -0.05 ↓ | 0.20 -0.03 ↓ |
| | Dim Reduction | | 0.36 0.08 ↑ | 0.22 -0.01 ↓ | 0.32 -0.06 ↓ | 0.27 -0.07 ↓ | 0.11 -0.05 ↓ | 0.12 -0.03 ↓ | 0.21 -0.06 ↓ | 0.23 -0.03 ↓ |
| | SBERT | | 0.38 0.06 ↑ | 0.27 -0.04 ↓ | 0.31 -0.08 ↓ | 0.29 -0.08 ↓ | 0.16 -0.08 ↓ | 0.14 -0.03 ↓ | 0.27 -0.08 ↓ | 0.26 -0.05 ↓ |
| | BPR | | 0.38 0.06 ↑ | 0.33 -0.03 ↓ | 0.34 -0.08 ↓ | 0.33 -0.08 ↓ | 0.19 -0.06 ↓ | 0.16 -0.04 ↓ | 0.33 -0.09 ↓ | 0.29 -0.05 ↓ |
| | Faiss | | 0.44 0.05 ↑ | 0.36 -0.03 ↓ | 0.40 -0.09 ↓ | 0.38 -0.09 ↓ | 0.22 -0.09 ↓ | 0.19 -0.04 ↓ | 0.34 -0.10 ↓ | 0.33 -0.06 ↓ |
| | ANCE | | 0.41 0.07 ↑ | 0.34 -0.03 ↓ | 0.41 -0.08 ↓ | 0.38 -0.09 ↓ | 0.23 -0.11 ↓ | 0.20 -0.04 ↓ | 0.37 -0.11 ↓ | 0.35 -0.06 ↓ |
| | TCT-COLBERT | | 0.44 0.04 ↑ | 0.41 -0.03 ↓ | 0.41 -0.08 ↓ | 0.39 -0.10 ↓ | 0.23 -0.09 ↓ | 0.20 -0.03 ↓ | 0.38 -0.12 ↓ | 0.35 -0.06 ↓ |
| | Aggretriever | | 0.42 0.08 ↑ | 0.41 -0.04 ↓ | 0.42 -0.10 ↓ | 0.40 -0.08 ↓ | 0.24 -0.09 ↓ | 0.24 -0.04 ↓ | 0.40 -0.12 ↓ | 0.36 -0.06 ↓ |
| | Arctic | | 0.43 0.10 ↑ | 0.49 -0.05 ↓ | **0.50** -0.12 ↓ | **0.54** -0.09 ↓ | **0.31** -0.12 ↓ | **0.35** -0.05 ↓ | **0.49** -0.13 ↓ | **0.44** -0.07 ↓ |
| Rerank | BM25 | TBERT | 0.40 0.07 ↑ | 0.40 -0.01 ↓ | 0.38 -0.07 ↓ | 0.38 -0.08 ↓ | 0.25 -0.07 ↓ | 0.19 -0.02 ↓ | 0.39 -0.10 ↓ | 0.34 -0.04 ↓ |
| | | NBoost | 0.41 0.08 ↑ | 0.44 -0.01 ↓ | 0.43 -0.09 ↓ | 0.44 -0.08 ↓ | 0.27 -0.09 ↓ | 0.22 -0.03 ↓ | 0.43 -0.12 ↓ | 0.38 -0.05 ↓ |
| | | MiniLM | 0.42 0.07 ↑ | 0.44 -0.01 ↓ | 0.42 -0.10 ↓ | 0.43 -0.09 ↓ | 0.26 -0.09 ↓ | 0.22 -0.04 ↓ | 0.42 -0.12 ↓ | 0.37 -0.05 ↓ |
| | | ELEC | 0.40 0.08 ↑ | 0.43 0.00 ↑ | 0.43 -0.07 ↓ | 0.43 -0.07 ↓ | 0.27 -0.08 ↓ | 0.22 -0.03 ↓ | 0.42 -0.11 ↓ | 0.37 -0.04 ↓ |
| | | MonoT5 | 0.42 0.08 ↑ | 0.45 0.00 ↑ | 0.45 -0.08 ↓ | 0.46 -0.07 ↓ | 0.30 -0.07 ↓ | 0.25 -0.02 ↓ | 0.44 -0.11 ↓ | 0.40 -0.04 ↓ |
| | ANCE | TBERT | 0.43 0.07 ↑ | 0.42 -0.03 ↓ | 0.41 -0.09 ↓ | 0.40 -0.09 ↓ | 0.26 -0.11 ↓ | 0.22 -0.03 ↓ | 0.40 -0.12 ↓ | 0.36 -0.06 ↓ |
| | | NBoost | 0.43 0.07 ↑ | 0.46 -0.03 ↓ | 0.47 -0.10 ↓ | 0.45 -0.10 ↓ | 0.28 -0.12 ↓ | 0.25 -0.04 ↓ | 0.44 -0.14 ↓ | 0.40 -0.06 ↓ |
| | | MiniLM | **0.46** 0.06 ↑ | 0.46 -0.03 ↓ | 0.46 -0.10 ↓ | 0.45 -0.11 ↓ | 0.27 -0.13 ↓ | 0.25 -0.05 ↓ | 0.43 -0.14 ↓ | 0.40 -0.07 ↓ |
| | | ELEC | 0.44 0.08 ↑ | 0.45 -0.02 ↓ | 0.46 -0.09 ↓ | 0.44 -0.09 ↓ | 0.27 -0.12 ↓ | 0.25 -0.04 ↓ | 0.43 -0.13 ↓ | 0.39 -0.06 ↓ |
| | | MonoT5 | 0.44 0.08 ↑ | 0.46 -0.02 ↓ | **0.50** -0.09 ↓ | 0.49 -0.08 ↓ | **0.31** -0.11 ↓ | 0.28 -0.04 ↓ | 0.45 -0.13 ↓ | 0.42 -0.06 ↓ |
| Rewrite | BM25 | HyDE | 0.31 0.14 ↑ | 0.49 0.06 ↑ | 0.32 0.03 ↑ | 0.29 0.02 ↑ | 0.25 0.10 ↑ | 0.21 0.09 ↑ | 0.33 0.03 ↑ | 0.31 0.07 ↑ |
| | | Q2D | 0.34 0.13 ↑ | 0.45 0.02 ↑ | 0.37 0.00 ↑ | 0.33 0.00 ↑ | 0.26 0.07 ↑ | 0.22 0.06 ↑ | 0.41 -0.01 ↓ | 0.34 0.04 ↑ |
| | | MuGI | 0.36 0.13 ↑ | **0.51** 0.03 ↑ | 0.39 0.00 ↑ | 0.35 -0.01 ↓ | 0.28 0.05 ↑ | 0.23 0.06 ↑ | 0.43 -0.01 ↓ | 0.36 0.04 ↑ |
| | Avg. | | 0.38 0.08 ↑ | 0.40 -0.01 ↓ | 0.39 -0.06 ↓ | 0.38 -0.06 ↓ | 0.23 -0.05 ↓ | 0.20 -0.01 ↓ | 0.38 -0.07 ↓ | 0.34 -0.03 ↓ |

## J.1 RETRIEVAL METRIC

**Hit@k.** Hit@k evaluates whether at least one of the top-k retrieved chunks contains the gold chunk(s). This metric is a binary metric that reflects success within the top-ranked candidates.

$$\text{Hit@k} = \mathbf{1}(\text{rank}^{rel} \leq k),$$

where $\text{rank}^{rel}$ denotes the rank of the first relevant chunk, and $\mathbf{1}$ is the indicator function that returns 1 if the condition holds, and 0 otherwise.

**nDCG@k.** nDCG@k accounts not only for the existence of gold chunks in the top-k retrieved chunks, but also their ranks. It assigns higher scores when relevant chunks appear earlier in the ranking, providing a position-sensitive measure of retrieval quality.

$$\text{nDCG@k} = \frac{\text{DCG@k}}{\text{IDCG@k}},$$

where

$$\text{DCG@k} = \sum_{i=1}^{k} \frac{2^{rel_i} - 1}{\log_2(i+1)},$$

$$\text{IDCG@k} = \sum_{i=1}^{k} \frac{2^{rel'_i} - 1}{\log_2(i+1)},$$

$rel_i$ denotes the relevance score of the $i$-th retrieved document, $rel'_i$ represents the relevance scores sorted in descending order, and IDCG@k is the ideal DCG@k, representing the maximum possible DCG@k under the optimal ranking.

Table 22: RAGAlign@10 score gains of BRIDGE for Qwen2.5-7B-Instruct.

| Version | BEIR | | ROBUSTQA | | | | | Avg. |
|---------|------|------|------|------|------|------|------|------|
| | MS | NQ | Life | Rec | Sci | Tech | Writ | |
| Original | 0.58 | 0.71 | 0.67 | 0.71 | 0.66 | 0.66 | 0.69 | 0.66 |
| BRIDGE | 0.83 | 0.80 | 0.71 | 0.75 | 0.75 | 0.76 | 0.73 | 0.76 |

Table 23: RAGAlign@10 score gains of BRIDGE for Gemma-2-9b-it.

| Version | BEIR | | ROBUSTQA | | | | | Avg. |
|---------|------|------|------|------|------|------|------|------|
| | MS | NQ | Life | Rec | Sci | Tech | Writ | |
| Original | 0.60 | 0.74 | 0.73 | 0.73 | 0.68 | 0.68 | 0.72 | 0.69 |
| BRIDGE | 0.85 | 0.86 | 0.76 | 0.78 | 0.76 | 0.76 | 0.75 | 0.78 |

## J.2 GENERATION METRIC

We employ GPT-4o-based binary evaluation to assess the correctness of generated answers against ground truth annotations. The evaluation prompt is structured to determine whether the predicted answer captures the core content of any ground truth answer, returning a binary True/False judgment. Table 25 presents the specific prompt template used for this evaluation.

## J.3 ALIGNMENT METRIC

We extend the RAGAlign metric from a binary to non-binary variables, enabling the computation of non-binary metrics such as nDCG@k and Recall@k.

**RAGAlign@k between Binary Variables.** When both retrieval and generation metrics are binary, we measure alignment as the proportion of queries where both components agree in their evaluation outcomes:

$$\text{RAGAlign@K} = \frac{1}{N} \sum_{i=1}^{N} \mathbf{1} \left[ \text{METRICR}_i = \text{METRICG}_i \right]$$

where $N$ denotes the total number of queries in the dataset. $\text{METRICR}_i \in \{0, 1\}$ is the binary retrieval outcome, and $\text{METRICG}_i \in \{0, 1\}$ is the binary generation outcome for the $i$-th query. $\mathbf{1}$ is the indicator function that outputs 1 if the condition is satisfied, and 0 otherwise. This metric captures how consistently retrieval and generation evaluations align.

**RAGAlign@k between Non-binary and Binary Variables.** Point-biserial correlation is a special form of Pearson correlation used when one variable is binary and the other is non-binary. This allows us to measure alignment between continuous retrieval metrics (*e.g.*, nDCG@k) and binary generation outcomes:

$$\text{RAGAlign@k} = \frac{\bar{R}_1 - \bar{R}_0}{S.E(R)} \cdot \sqrt{\frac{n_1 n_0}{n(n-1)}}$$

where $\bar{R}_1$ and $\bar{R}_0$ are the mean METRICR for METRICG=1 and METRICG=0 respectively, $S.E(R)$ is the standard error of METRICR, and $n_1$, $n_0$, $n$ are the corresponding sample sizes. The point-biserial correlation can be computed as the Pearson correlation between a non-binary variable and a binary variable coded as 0 and 1.

**Result.** To investigate whether the RAGAlign@10 gains on BRIDGE generalize to other LLM generators, we additionally evaluate the retrieval–generation alignment rate on two further generators, Qwen2.5-7B-Instruct and Gemma-2-9b-it in Table 22 and Table 23. Both models exhibit the similar trend as Llama-3.1-8B-Instruct, demonstrating that the observed gains are not tied to a single generator configuration. We will include these results in the revised version.

Table 21: RAGAlign@10 score gains over seven subsets of BRIDGE measured by nDCG@10

| Version | BEIR | | ROBUSTQA | | | | | Avg. |
|---------|------|------|------|------|------|------|------|------|
| | MS | NQ | Life | Rec | Sci | Tech | Writ | |
| Original | 0.16 | 0.38 | 0.39 | 0.43 | 0.28 | 0.34 | 0.33 | 0.33 |
| BRIDGE | 0.38 | 0.51 | 0.45 | 0.49 | 0.45 | 0.51 | 0.42 | 0.46 |

Table 21 demonstrates the nDCG@10-based RAGAlign@10 scores comparing BRIDGE to the original benchmark. Consistent performance improvements are observed across all datasets, with MS MARCO showing the most substantial gain from 0.16 to 0.38. The RobustQA subdomains exhibit more moderate, but positive improvements across all areas. On average, the RAGAlign@10 score improved from 0.33 to 0.46, confirming that chunk label correction significantly enhances the accuracy of retrieval performance evaluation.

Table 24: Details of the model checkpoint and hardware used for implementation.

| Model Type | Model Name | Hugging Face Checkpoints & Official API Version | Hardware & Precision |
|---|---|---|---|
| Proprietary LLMs | GPT-4o-mini
GPT-4o | gpt-4o-mini-2024-07-18
gpt-4o-2024-08-06 | OpenAI API & Default settings
OpenAI API & Default settings |
| Open-source LLMs | Llama3.1-8B-Inst.
Llama3.3-70B-Inst.
Qwen2.5-72B-Inst. | meta-llama/Llama-3.1-8B-Instruct
meta-llama/Llama-3.3-70B-Instruct
Qwen/Qwen2.5-72B-Instruct | NVIDIA L40S 48GB (×1) & BF16
NVIDIA L40S 48GB (×4) & BF16
NVIDIA L40S 48GB (×4) & BF16 |

### J.4 GENERATION PHASE

Our generation setup employs a consistent approach across all systems to isolate the impact of retrieval quality. We utilize a single large language model `Llama-3.1-8B-Instruct` with a standardized prompt template that incorporates the top-10 retrieved chunks as context. This uniform generation configuration ensures that performance variations can be directly attributed to differences in retrieval quality rather than generation capabilities. The specific prompt structure enforces strict JSON formatting and prevents the model from incorporating external knowledge beyond the provided context. For the generation prompt, see Table 26.

## K  LLM IMPLEMENTATION DETAILS

Table 24 provides comprehensive details of model versions and hardware specifications used in our experiments. Open-source models are sourced from Hugging Face with specific checkpoint versions, and we detail the exact hardware configuration, GPU memory and precision settings used for each model deployment. Proprietary models are accessed via official APIs including OpenAI API and AWS Bedrock with documented API versions and parameters.

Table 25: Prompt for LLM-based binary evaluation.

Your task is to evaluate the correctness of the PREDICTED ANSWER based on the GT ANSWERs.

### Instructions:
- Read the QUERY and then compare the GT ANSWERs and the PREDICTED ANSWER.
- Check if the PREDICTED ANSWER includes any of the core content of the GT ANSWERs.
- If there are multiple GT ANSWERS and the PREDICTED ANSWER includes the core content of at least one of them, output "True".

### QUERY:
{query}

### GT ANSWERs:
{ground truths}

### PREDICTED ANSWER:
{prediction}

### Strictly output True or False

Table 26: Prompt for RAG answer generation.

Answer the given QUERY only using the information provided in the Multiple CONTEXTs. Do not include any assumptions, general knowledge, or information not found in the Multiple CONTEXTs.

QUERY: {query}
Multiple CONTEXTs: {retrieved documents}

Do not provide any explanation or additional text.

Respond strictly in the following JSON format:

- If **no relevant information** is found in CONTEXTs:
{"Answer": "No relevant information found."}

- If **relevant information** exists in CONTEXTs, answer in short form:
{"Answer": "your answer"}

Table 27: Prompt for initiating the first-round debate, containing both initial stances.

**System Instruction**

You are {agent1}, and your task is to determine whether the given chunk FULLY answer to the query with AT LEAST ONE of the answers. There is also other agent, {agent2}, assigned the same task as you. You are provided query, its answers, and chunk in `<query></query>` tags, `<ans></ans>` tags, and `<doc></doc>` tags respectively. There can be multiple answers to the query, each listed with an ordered number. You can also access to the discussion history with {agent2} in `<history></history>` tags. You can refer your previous argument and {agent2} argument. There are also important guidelines to help your decision in `<guide></guide>` tags and keep that it mind for whole discussion process.

**Target Input**

```
<query>
```
**{query}**
```
</query>
<answer>
```
**{answer}**
```
</answer>
<doc>
```
**{chunk}**
```
</doc>
<history>
```
{agentA}: Yes. The chunk contains complete information to construct at least one answer to the target query.
{agentB}: No. The chunk does not contain complete information for any answer to the target query.
```
</history>
```

**Debate Guidelines**

You have to carefully read the query and its answers and fully understand the core content of each answer. Afterthat, read the given chunk and fully understand it. With your understanding of the chunk and answers, you have to analyze the discussion history and engage critically with the discussion. Then, you have to think critically whether the chunk can fully justify any of answers with all of the guidelines in `<guide></guide>`.
`<guide>`
1. The chunk has the same scope as the scope of the answer. The chunk with a specific example cannot justify a general definition, and vice versa.
2. The chunk must contain the key contents of the answer, and present a consistent context without contradiction with the answer.
3. The chunk must directly provide the answer, without implications, oppositional logic or common-sense reasoning.
4. The chunk must express the same concept with the same intent, scope and practical instruction. A simple match on theme or terminology is not sufficient.
5. Base your decision solely on whether the chunk provides complete information for the answers, regardless of any additional information.
6. Do not mark the answer as supported based on surface-level similarity, for example both mention same word, without checking for matching intent and guidance.
7. Do not assess answer correctness, relevance to query, or consistency between answers.
8. Each answer must be evaluated independently.
`</guide>`
Explain your reasoning process step by step, including how you interpreted the chunk, and engaged with the other agent's position. If applicable, include a reference sentence from the chunk to support your reasoning. Conclude with a clear justification of your final decision. Determine whether the given chunk is fully supporting any of the answers in "response"
- yes: The chunk contains complete information to construct at least one answer to the target query.
- no: The chunk does not contain complete information for any answer to the target query.

**Output Template**

Provide your output as a properly formatted JSON object. No additional explanation.
{ "reference": ["reference sentence 1 if exists", ...], "reason": "Explain your reasoning process step by step. Conclude with a clear justification of your final decision. (Max 100 words)", "response": "yes or no" }

Table 28: Prompt for the subsequent-round debate, incorporating the discussion history from the previous round.

**System Instruction**

You are {agent1}, and your task is to determine whether the given chunk FULLY answer to the query with AT LEAST ONE of the answers. There is also other agent, {agent2}, assigned the same task as you. You are provided query, its answers, and chunk in `<query></query>` tags, `<ans></ans>` tags, and `<doc></doc>` tags respectively. There can be multiple answers to the query, each listed with an ordered number. You can also access to the discussion history with {agent2} in `<history></history>` tags. You can refer your previous argument and {agent2} argument. There are also important guidelines to help your decision in `<guide></guide>` tags and keep that it mind for whole discussion process.

**Target Input**

```
<query>
```
**{query}**
```
</query>
<answer>
```
**{answer}**
```
</answer>
<doc>
```
**{chunk}**
```
</doc>
<history>
```
**{agentA}: {agentA_reasoning}**
**{agentB}: {agentB_reasoning}**
```
</history>
```

**Debate Guidelines**

You have to carefully read the query and its answers and fully understand the core content of each answer. Afterthat, read the given chunk and fully understand it. With your understanding of the chunk and answers, you have to analyze the discussion history and engage critically with the discussion. Then, you have to think critically whether the chunk can fully justify any of answers with all of the guidelines in `<guide></guide>`.
```
<guide>
```
1. The chunk has the same scope as the scope of the answer. The chunk with a specific example cannot justify a general definition, and vice versa.
2. The chunk must contain the key contents of the answer, and present a consistent context without contradiction with the answer.
3. The chunk must directly provide the answer, without implications, oppositional logic or common-sense reasoning.
4. The chunk must express the same concept with the same intent, scope and practical instruction. A simple match on theme or terminology is not sufficient.
5. Base your decision solely on whether the chunk provides complete information for the answers, regardless of any additional information.
6. Do not mark the answer as supported based on surface-level similarity, for example both mention same word, without checking for matching intent and guidance.
7. Do not assess answer correctness, relevance to query, or consistency between answers.
8. Each answer must be evaluated independently.
```
</guide>
```
Explain your reasoning process step by step, including how you interpreted the chunk, and engaged with the other agent's position. If applicable, include a reference sentence from the chunk to support your reasoning. Conclude with a clear justification of your final decision. Determine whether the given chunk is fully supporting any of the answers in "response"
- yes: The chunk contains complete information to construct at least one answer to the target query.
- no: The chunk does not contain complete information for any answer to the target query.

**Output Template**

Provide your output as a properly formatted JSON object. No additional explanation.
{ "reference": ["reference sentence 1 if exists", ...], "reason": "Explain your reasoning process step by step. Conclude with a clear justification of your final decision. (Max 100 words)", "response": "yes or no" }

Table 29: Adjudicator prompt to resolve the disagreement between two agents.

**System Instruction**

You are an ADJUDICATOR tasked with making the final determination on chunk-answer-query alignment. You are given a query, answers, a chunk, and discussion history between two evaluator agents with opposing views. Determine whether the chunk contains sufficient information to fully substantiate AT LEAST ONE of the candidate answers to the given query.
- The answers are proposed responses to the query.
- The chunk should provide complete evidentiary support for constructing/validating at least one answer.
- You are NOT evaluating answer quality, only answer-query grounding.

**Target Input**

```
<query>
{query}
</query>
<answer>
{answer}
</answer>
<doc>
{chunk}
</doc>
<history>
{history}
</history>
```

**Adjudication Guidelines**

You have to carefully read the query and its answers and fully understand the core content of each answer. Afterthat, read the given chunk and fully understand it. Summarize both agents' positions and evaluate their reasoning quality. Assess how well each agent followed the evaluation criteria below in `<guide></guide>`. Then, you have to think critically whether the chunk can fully justify any of the answers with all of the guidelines in `<guide></guide>`.
`<guide>`
You must stick to all of the guidelines below for your decision:
1. The chunk has the same scope as the scope of the answer. The chunk with a specific example cannot justify a general definition, and vice versa.
2. The chunk must contain the key contents of the answer, and present a consistent context without contradiction with the answer.
3. The chunk must directly provide the answer, without implications, oppositional logic or common-sense reasoning.
4. The chunk must express the same concept with the same intent, scope and practical instruction. A simple match on theme or terminology is not sufficient.
5. Base your decision solely on whether the chunk provides complete information for the answers, regardless of any additional information.
6. Do not mark the answer as grounded based on surface-level similarity, for example both mention the same word, without checking for matching intent and guidance.
`</guide>`
Explain your reasoning process step by step, including how you interpreted the chunk, engaged with both agents' arguments, and applied the evaluation guidelines. Conclude with a clear justification of your final decision. Determine whether the given chunk is fully substantiating any of the answers in "response".
- yes: The chunk contains complete information to construct at least one answer to the target query.
- no: The chunk does not contain complete information for any answer to the target query.

**Output Template**

Provide your output as a properly formatted JSON object. No additional explanation.
{ "reference": ["reference sentence 1 if exists", ...], "reason": "Explain your reasoning process step by step. Conclude with a clear justification of your final decision. (Max 100 words)", "response": "yes or no" }

