# OpenReview forum: "Completing Missing Annotation: Multi-Agent Debate for Accurate and Scalable Relevant Assessment for IR Benchmarks"
_ICLR.cc/2026/Conference — ICLR 2026 Poster_

### Official Review · Reviewer_58tj · 2025-10-26

**Soundness:** 3
**Presentation:** 2
**Contribution:** 3
**Rating:** 6
**Confidence:** 5

**Summary:**

The paper aims to address missing relevance annotations (“holes”) in IR benchmarks that distort evaluation and hinder RAG alignment. It proposes DREAM, a multi-agent debate framework where two LLMs take opposing stances (Relevant vs. Irrelevant) and iteratively critique each other until reaching consensus or escalating to humans. Using this method, the authors rebuild benchmarks into BRIDGE, filling large annotation gaps and revealing that many retrieval improvements were previously underestimated. The results show DREAM effectively automates reliable labeling and that completing these “holes” leads to more accurate and faithful IR and RAG evaluations.

**Strengths:**

1. The paper highlights an often-overlooked issue — that retrieval performance itself has been misjudged because of annotation holes in IR benchmarks. This brings a fresh perspective to understanding the mismatch between retrieval and generation performance.

2. I really like how the multi-agent debate setup helps counter problems like single-model overconfidence or failed confidence-based escalation, where easy cases get escalated and truly hard ones slip through.

**Weaknesses:**

One thing I wonder is whether the framework considers multi-hop reasoning cases — where the answer can only be inferred by combining multiple chunks. For example, if one passage says “Jack’s mother is Jane” and another says “Jane’s husband is Bob,” neither chunk alone directly answers “Who is Jack’s father?”, but together they clearly provide the evidence. It’d be interesting to see how DREAM handles or could be extended to capture this kind of compositional relevance.

**Questions:**

1. By default, the debate runs with two rounds of alternating arguments (R=2). I’m curious whether the authors explored what happens with more rounds — for instance, does extending the debate yield deeper reasoning or just redundancy? Some analysis on how performance scales with R could offer useful insights into the debate dynamics.

2. Since both agents (m₁ and m₂) are initialized from the same Llama3.3-70B-Instruct model, I wonder if there’s any risk of “shared bias” or reinforcement from their common pretraining knowledge, even though they take opposing stances. Would using models from different sources (e.g., different architectures or training data) introduce useful diversity, or might it hurt the framework’s stability and its strong human-level accuracy?

---

> ### Author Response · Authors · 2025-11-25
> **Response to Reviewer 58tj (1/3)**
>
> ---
>
> We sincerely appreciate for the reveiwer's thoughtful comments. We have carefully reviewed your comments and provide detailed responses to each point below. We hope our rebuttal adequately addresses the raised concerns, and all discussed revisions will be incorporated into the updated manuscript.
>
> ---
>
> ## W1. Multi-hop Reasoning Cases
> > *One thing I wonder is whether the framework considers multi-hop reasoning cases — where the answer can only be inferred by combining multiple chunks. For example, if one passage says “Jack’s mother is Jane” and another says “Jane’s husband is Bob,” neither chunk alone directly answers “Who is Jack’s father?”, but together they clearly provide the evidence. It’d be interesting to see how DREAM handles or could be extended to capture this kind of compositional relevance.*
>
> We appreciate the reviewer's thoughtful question on how `DREAM` handles multi-hop evidence aggregation. We agree that multi-hop reasoning is an important aspect of relevance assessment.
>
> The key insight is that multi-hop queries can be viewed as equivalent to multiple single-hop subqueries connected through logical reasoning. Recent work has demonstrated that complex queries can be effectively decomposed into simpler subqueries [1, 2]. Since `DREAM` is designed to handle single-hop relevance assessment effectively, extending it to multi-hop scenarios becomes straightforward through query decomposition.
>
> In the reviewer's example, the multi-hop query "Who is Jack's father?" can be decomposed into single-hop sub-queries such as "Who is Jack's mother?" and "Who is Jane's husband?". Notably, in our evaluation setting, we have access to both the query and its ground-truth answer during query decomposition, enabling us to generate logically coherent subqueries that capture the necessary reasoning steps. `DREAM` can then evaluate relevance at the subquery level, assessing whether chunks satisfy these individual subqueries. By aggregating the relevance assessments across all subqueries, we can naturally extend `DREAM` to handle multi-hop reasoning without modifying its core evaluation mechanism.
>
> This extension requires minimal architectural changes, as we simply apply our existing single-hop methodology to each decomposed subquery and integrate the results. We believe this is a valuable direction for future work.
>
> [1] Khot et al., "Decomposed Prompting: A Modular Approach for Solving Complex Tasks," ICLR 2023.\
> [2] Zhou et al., "Least-to-Most Prompting Enables Complex Reasoning in Large Language Models," ICLR 2023.
>
> ---

---

> ### Author Response · Authors · 2025-11-25
> **Response to Reviewer 58tj (2/3)**
>
> ---
>
> ## Q1. Multiple-Round Debate (1/2)
>
> > ***Q1-1**: By default, the debate runs with two rounds of alternating arguments (R=2). I’m curious whether the authors explored what happens with more rounds - for instance, does extending the debate yield deeper reasoning or just redundancy?*
>
> ### Qualitative analysis on Multiple Rounds
> We observe that extending the debate has different effects depending on the case. When disagreement persists, the reasoning becomes deeper as agents critically evaluate and elaborate on each other’s arguments. However, once consensus is reached, later rounds add only surface-level changes with no new evidence, leading to redundancy rather than deeper reasoning.
>
> **Example: Deepening Reasoning in Cases with Persist Disagreement**
>
> As the debate progresses, the reasoning becomes more elaborate, with agents critically evaluating and systematically responding to one another’s arguments while adding their own interpretations.
>
>     Example
>     <R1>
>     - AgentA(Relevant): "The document provides a range of 70-130 mg/dL, which includes the answer range of 70-100 mg/dL, thus fully justifying answer [0]."
>     - AgentB(Irrelevant): "The document provides a range that includes the answer, but also introduces conflicting information."
>
>     <R2>
>     - AgentA(Relevant): "I interpreted the document as providing a range that includes the answer. AgentB's concern about conflicting information is noted, but the document directly provides a range that encompasses the answer. The document's scope matches the answer's scope."
>     - AgentB(Irrelevant): "I interpreted the document as providing a range that includes the answer. AgentA's position is that this range fully justifies answer [0]. However, I disagree due to conflicting information in the document."
>
>     <R3>
>     - AgentA(Relevant): "I interpreted the document as providing a range that includes the answer. The document's scope matches the answer's scope, and it directly provides a range that encompasses the answer. AgentB's concern about conflicting information is noted, but the document's range of 70-130 mg/dL includes the answer's range of 70-100 mg/dL. I evaluated the answer independently and found that the document provides complete information for answer [0]."
>     - AgentB(Irrelevant): "I interpreted the document as providing conflicting information regarding the ideal fasting glucose level. The document mentions a range of 70-130 mg/dL recommended by the American Diabetes Association, which includes the answer [0] 70-100 mg/dL. However, it also mentions that some sources consider anything over 90 as not good, which introduces inconsistency. Engaging with AgentA's position, I understand their point that the document provides a range that encompasses the answer, but I disagree due to the conflicting information. The document does not directly provide a consistent answer without implications."
>
> **Example: Limited Reasoning Gain After Consensus**
>
> Once consensus is reached, the arguments across subsequent rounds change only in surface-level. The agents' justifications show minor phrasing differences but converge to similar reasoning.
>
>     Example
>     <R2>-Consensus
>     - AgentA(Irrelevant): "I interpreted the document as providing a specific example of calculating boat transport cost. AgentB argued it doesn't directly state the general cost range, but I think the example implies a range. However, guideline 3 states the document must directly provide the answer without implications."
>     - AgentB(Irrelevant): "I interpreted the document as providing a specific example, not a general cost range. AgentA's position is incorrect as it implies a calculation method, not a direct answer. The document does not directly state the general cost range."
>
>     <R3>-Consensus
>     - AgentA(Irrelevant): "The document provides a specific example, not a general cost range, and doesn't directly state the answer without implications."
>     - AgentB(Irrelevant): "The document provides a specific example, not a general cost range. It implies a calculation method, not a direct answer, contradicting guideline 3."
>
>     <R4>-Consensus
>     - AgentA(Irrelevant): "The document provides a specific example, not a general cost range, and implies a calculation method, not a direct answer, contradicting guideline 3."
>     - AgentB(Irrelevant): "The document provides a specific example, not a general cost range. It implies a calculation method, not a direct answer, contradicting guideline 3."
>
> ---

---

> ### Author Response · Authors · 2025-11-25
> **Response to Reviewer 58tj (3/3)**
>
> ---
>
> ## Q1. Multiple-Round Debate (2/2)
>
> > ***Q1-2:** Some analysis on how performance scales with R could offer useful insights into the debate dynamics.*
> ### Quantitative Analysis across Multiple Rounds
>
> We additionally provide a quantitative performance analysis to examine how labeling performance changes as the number of debate rounds R increases. We continue the debate for multiple rounds even when earlier rounds already reached consensus, and compute performance solely on the consensus cases at each round.
>
> | Round | Recall (Irrelevance) | Recall (Relevance) |  bAcc | Escalation Ratio |
> |-----|---|-----|-----|-----|
> |   R1  |  92.7%  | 99.0% | 95.8% | 10.5% |
> |   R2  |  93.6%  | 98.1% | 95.8% |  5.3% |
> |   R3  |  93.2%  | 93.6% | 93.4% |  2.0% |
> |   R4  |  93.5%  | 92.6% | 93.0% |  1.1% |
> |   R5  |  93.5%  | 92.2% | 92.9% |  1.1% |
>
> **Improvement from R1 to R2:** The labeling accuracy (bAcc) reaches its highest point at Round 2 (95.8%), and this round simultaneously achieves the lowest escalation ratio of 5.3%, because R2 represents the ideal balance between resolving meaningful disagreements in R1 and preserving correct early consensus.
>
> **Breakdown Beyond R2:** However, once the debate proceeds beyond R2, this balance seems to break down. Ambiguous disagreement cases and some previously correct consensuses begin to collapse toward the "irrelevant" label. This shift occurs because repeated argument exchanges amplify conservative tendencies, dilute the distinct evidence introduced earlier, and increasingly push agents to avoid false positives by defaulting to irrelevance. Consequently, relevance recall drops gradually, bAcc declines, and the lower escalation ratio in later rounds reflects incorrect convergence.
>
> ---
>
> ## Q2. Different Model Families
> > *Since both agents (m₁ and m₂) are initialized from the same Llama3.3-70B-Instruct model, I wonder if there’s any risk of “shared bias” or reinforcement from their common pretraining knowledge, even though they take opposing stances.*
> >
> > *Would using models from different sources (e.g., different architectures or training data) introduce useful diversity, or might it hurt the framework’s stability and its strong human-level accuracy?*
>
> We appreciate to the reviewer's concern about shared bias from the same model and suggestion for utilizing models from different sources.
>
> **Analysis 1: Experiment on Model Families**
>
> To investigate whether shared bias exists in the 2-Llama baseline and whether heterogeneous models can mitigate it, we conducted three more experiments:
>
> - (1) 2-Llama: Both agents use Llama3.3-70B-Instruct (our default setup)
> - (2) 2-Qwen: Both agents use Qwen2.5-72B-Instruct
> - (3) Llama(Relevant) & Qwen(Irrelevant): Agent initialized with "relevant" stance uses Llama, and agent with "irrelevant" stance uses Qwen
> - (4) Qwen(Relevant) & Llama(Irrelevant): The reverse of (3)
>
> We include configurations (3) and (4) to isolate the effect of model choice from stance assignment, since each model may perform differently depending on which stance it defends. All experiments use the same 728 samples using same setup as in Section 3.2.1, and performance is computed only on consensus cases (non-escalated).
>
> | Method | Recall (Irrelevance) |  Recall (Relevance) | bAcc | Escalation Ratio |
> | ----- | ----- | ----- | ----- | ----- |
> | `2-Llama` | 91.9% | 98.4% | 95.2% | 3.5% |
> | `2-Qwen`  | 95.1% | 82.8% | 89.0% | 2.6% |
> | `Llama(Relevant) & Qwen(Irrelevant)` | 95.8% | 88.7% | 92.3% | 6.6% |
> | `Qwen(Relevant) & Llama(Irrelevant)` | 96.0% | 91.4% | 93.7% | 6.5% |
>
>
> The 2-Llama baseline shows higher relevant recall than irrelevant recall, while the 2-Qwen baseline shows the opposite pattern. This indicates that Llama is biased toward predicting relevance and Qwen is biased toward predicting irrelevance. The heterogeneous setup reduces these directional biases, achieving higher irrelevant recall than 2-Llama and higher relevant recall than 2-Qwen. However, its labeling accuracy (bAcc) remains lower than 2-Llama because Qwen contributes weaker judgments overall. This implies that heterogeneous debate is effective only when all participating models are individually strong.
>
> In conclusion, diversity across model families does not always provide benefits. It helps reduce bias, but it can degrade overall quality if one family underperforms. Selecting heterogeneous models therefore requires careful optimization.
>
> ---
>
> If there are any aspects of our response that remain unclear or could benefit from further clarification, we would be more than happy to address them.
>
> ---

---

> ### Comment · Reviewer_58tj · 2025-11-26
>
> Thank you for the additional explanations and comparative experiments. They clarified the framework’s behavior and helped me better understand the issues raised. However, considering the overall impression, I will reserve my original score. I encourage the authors to integrate these analyses and examples directly into the main manuscript, as they would strengthen the paper’s clarity and completeness.

---

> ### Author Response · Authors · 2025-11-26
> **Response to Reviewer Feedback**
>
> We are glad to hear that our additional explanations and comparative analyses helped clarify the framework and addressed the concerns you raised. As you suggested, we will incorporate these analyses and examples directly into the main manuscript and upload the revised version within the rebuttal period. If you have any further points to discuss, please feel free to let us know.
>
> Sincerely, authors

---

### Official Review · Reviewer_M8G8 · 2025-10-27

**Soundness:** 4
**Presentation:** 3
**Contribution:** 3
**Rating:** 6
**Confidence:** 4

**Summary:**

The focus of the paper is on IR evaluation, where the authors propose an LLM multi-round debate solution. The solution is motivated by the fact that current IR benchmarks have "holes", i.e. data that is not labelled as relevant but might be relevant. Past work has used LLMs for relevance assessment but used single judge, which suffers from overconfidence. The authors point out that this hurts the performance of hybrid systems that send uncertain cases to humans: to do this well, models must be calibrated/ambiguity-aware.

The paper introduces DREAM to use adversarial debate to decide when to escalate to human judgment. The debate is based on conflicting stances, only escalating when agents agree. They introduce a benchmark based on existing datasets, using DREAM to reduce evaluation holes.

The paper finds that DREAM reduces the escalation ratio while improving accuracy on the evaluation set. Moreover, DREAM can be used to refine real-world IR datasets and fill holes, lowering the human cost by only human-annotating uncertain cases. The refined real-world IR data is used to evaluate systems for RAG, with improvements to a RAG model.

**Strengths:**

- Strong intrinsic results: the paper shows that DREAM greatly reduces the amount of human annotation needed for the same accuracy
- Downstream utility: the paper demonstrates the utility of DREAM on augmenting a real IR dataset, where human annotation cost would be high with baseline approaches.
- Human evaluation: the evaluation set is vetted by human expert annotators
- Baselines: the method compares to LLM-only and confidence-based baselines that support its claims.

**Weaknesses:**

- No discussion of latency cost: compared to LLM-as-judge baselines, this debate approach is much more expensive, requiring multiple model calls across multiple rounds. While the method saves human cost, the trade-off here should be discussed.
- Novelty/major missing related work section: the paper is missing most related work on multi-agent debate, including https://arxiv.org/abs/2309.13007, https://arxiv.org/abs/2305.14325, https://arxiv.org/abs/2504.13079 which have covered stances, divergent thinking, and evidence conflict. The application of debate to IR evaluation seems novel but the debate methods themselves less so.
- Dataset size: the size of the evaluation dataset seems quite small compared to the original datasets (BEIR and RobustQA) which have tens of thousands of examples.
- This paper would benefit from interaction with the extensive literature on selective prediction, which has the same motivation (e.g. https://aclanthology.org/2022.findings-acl.158/, https://arxiv.org/abs/2303.16857, https://arxiv.org/abs/2402.15610)
- Comparison to LARA would be easier to understand as a plot with different confidence thresholds/escalation ratios on the x axis

**Questions:**

- Order-dependence: LLMs are often prone to sycophancy and there's a documented trend of models picking whichever answer goes first. Does it matter which agent goes first? It would be good to have an ablation showing stance order doesn't matter.
- How do you evaluate consensus? Are models instructed to format answers in an extractable way?
- What was the reason for human disagreement on human eval?
- Have you thought about distillation? Past work, e.g. https://arxiv.org/abs/2402.01620 distills multi-agent interactions into a single model. In principle, you could distill the better classifications from the DREAM debate pipeline into a single judge model that at test time produces a single decision.

I'm willing to increase my score if these weaknesses/questions can be addressed.

Other minor points:
- L174: the opponents -> the opponent's
- L092-094: the common explanation...: this needs a citation. I believe another more common explanation for this is that the models are ignoring their contextual knowledge in favor of the parametric knowledge (https://aclanthology.org/2021.emnlp-main.565/, https://aclanthology.org/2022.emnlp-main.146/, https://aclanthology.org/2023.findings-emnlp.968/).

---

> ### Author Response · Authors · 2025-11-25
> **Response to Reviewer M8G8 (1/5)**
>
> We sincerely appreciate the reviewer's careful reading of our paper and the detailed comments, including the minor points, which have helped improve our work. We have carefully considered the feedback and respond to each issue individually below. All discussed revisions will be incorporated into the updated manuscript.
>
> ---
>
> ## W1. Latency Cost
> > *No discussion of latency cost: compared to LLM-as-judge baselines, this debate approach is much more expensive, requiring multiple model calls across multiple rounds. While the method saves human cost, the trade-off here should be discussed.*
>
> We acknowledge your concern about trade-off discussion. To address this, we explicitly compute the cost and latency per sample (i.e., a single {query–chunk–answer} triplet) and compare our methods (`DREAM`) against both `Human-Only` and `LLMJudge`.
>
> The "API Generation/Human Cost" column in the table below reports the human labor cost for `Human-Only` and API cost (based on input and output token costs under Deep Infra pricing) for both `LLMJudge` and `DREAM`. Latency is measured in two settings: "Non-parallel latency" corresponds to a single API session, where we sum the generation times of all participating agents. In contrast, "parallel latency" corresponds to multiple API sessions, measured as the wall-clock time when all agents generate their outputs concurrently. The reported results are averaged over 147 samples and we used Llama-3.3-70B-Instruct for `LLMJudge` and `DREAM`.
>
> | Method | Human or API Cost ($) |  Non-parallel Latency (sec) | Parallel Latency (sec) | bAcc (%) | Escalation Ratio (%)|
> | --- | --- | --- | --- | --- | --- |
> | `Human-Only` | 0.1 | 30 | 30 | 93.8 | 100.0 |
> | `LLMJudge` | 0.000066 | 1.292 | 1.292 | 73.9 | 0.0 |
> | `DREAM` | 0.000470 | 8.583 | 4.291 | 95.2 | 3.5 |
>
> `Human-Only` delivers strong labeling accuracy (bAcc), but this performance comes with very high human cost and latency. On the other hand, `LLMJudge` minimizes the cost and latency but its bAcc is significantly lower than `Human-Only`, indicating that it struggles to reliably identify relevance labels. In contrast, `DREAM` achieves even higher bAcc while being 200x cheaper and 3.5x--7.0x faster than `Human-Only`. This demonstrates that `DREAM` achieves a favorable balance between cost and performance, enabling scalable high-quality annotation that would be infeasible with human annotators alone.

---

> ### Author Response · Authors · 2025-11-25
> **Response to Reviewer M8G8 (2/5)**
>
> ---
>
> ## W2. Missing Related Work
> > ***W2-1:** Novelty/major missing related work section: the paper is missing most related work on multi-agent debate, including https://arxiv.org/abs/2309.13007, https://arxiv.org/abs/2305.14325, https://arxiv.org/abs/2504.13079 which have covered stances, divergent thinking, and evidence conflict.*
>
> We appreciate the reviewer for identifying the relevant related work and for providing several valuable references. We will expand the related work section to include recent multi-agent approaches and clarify how our framework differs from them, as shown below.
>
> **Multi-agent Debate.** Over a single-agent setup, recent studies have explored utilizing multiple agents with distinct roles that collaborate [1, 2]. In particular, multi-agent debate has been widely examined, where agents share their answers and rationales, critique one another, and iteratively refine their outputs over multiple rounds [3, 4]. Such debate-style collaboration has been applied to improve LLM and RAG answer quality [5, 6, 7], to handle knowledge conflicts in RAG [8], and to enhance response evaluation [9] and data annotation [10]. While existing work focuses primarily on AI-AI collaboration, we instead leverage multi-agent debate on AI-Human collaboration for IR relevance assessment. Our framework uses multi-agent debate not only to produce high-quality annotations, but also to determine when to escalate uncertain cases to humans and to improve human judgments through providing debate histories.
>
> [1] Chang et al., "MAIN-RAG: Multi-Agent Filtering Retrieval-Augmented Generation", ACL 2024.\
> [2] Shen et al., "Small LLMs Are Weak Tool Learners: A Multi-LLM Agent", EMNLP 2024.\
> [3] Xiong et al., "Examining Inter-Consistency of Large Language Models Collaboration: An In-depth Analysis via Debate", EMNLP 2023.\
> [4] Liang et al., "Encouraging Divergent Thinking in Large Language Models through Multi-Agent Debate", EMNLP 2024.\
> [5] Du et al., "Improving Factuality and Reasoning in Language Models through Multiagent Debate", ICML 2023.\
> [6] Chen et al., "ReConcile: Round-Table Conference Improves Reasoning via Consensus among Diverse LLMs", ACL 2024.\
> [7] Khan et al., "Debating with More Persuasive LLMs Leads to More Truthful Answers", ICML 2024.\
> [8] Wang et al., "Retrieval-Augmented Generation with Conflicting Evidence", COLM 2025.\
> [9] Chan et al., "ChatEval: Towards Better LLM-based Evaluators through Multi-Agent Debate", ICLR 2023.\
> [10] Tseng et al., "Evaluating Large Language Models as Expert Annotators", COLM 2025.
>
> > ***W2-2:** The application of debate to IR evaluation seems novel but the debate methods themselves less so.*
>
> We agree that our debate mechanism may appear less novel relative to prior multi-agent debate studies. However, our contribution lies not in introducing a new debate algorithm, but in fundamentally redefining how debate is applied and to what end. Prior debate-based studies do not address relevance assessment in IR; they focus on AI–AI collaboration to improve model performance, such as refining answers or enhancing response quality, where debate remains entirely within the model ecosystem and is evaluated only through downstream accuracy.
>
> In contrast, our work adapts multi-agent debate to AI–Human collaborative relevance assessment, which demands different design principles. Our framework uses debate to (1) generate high-quality IR labels, (2) detect uncertain cases for human escalation, (3) provide debate histories that improve human judgment, and (4) expand annotation capacity through hybrid AI–Human workflows. Thus, although the debate procedure resembles prior work, its purpose, usage, and interaction paradigm are entirely different. We believe this shift from AI–AI performance boosting to AI–Human collaborative annotation constitutes the core novelty of our approach.
>
> ---

---

> ### Author Response · Authors · 2025-11-25
> **Response to Reviewer M8G8 (3/5)**
>
> ---
>
> ## W3. Response to Dataset Size Concern
> > *Dataset size: the size of the evaluation dataset seems quite small compared to the original datasets (BEIR and RobustQA) which have tens of thousands of examples.*
>
> We agree that the current BRIDGE dataset is smaller than the full BEIR and RobustQA corpora. Importantly, however, the goal of our paper is not to replace these large benchmarks, but to evaluate the effectiveness of the DREAM annotation framework itself. For this purpose, a moderately sized and carefully curated dataset is sufficient to assess whether DREAM can (1) generate reliable relevance labels and (2) minimize human annotation effort.
>
> Moreover, DREAM proves highly cost-efficient: only 3.5% of examples require human intervention, implying that large-scale annotation (e.g., tens of thousands of queries) could be achieved at only a very small fraction of the cost of full manual labeling while maintaining comparable quality. Thus, While BRIDGE is intentionally modest in size for evaluation,  the scalability and cost-effectiveness demonstrated by DREAM show that the framework can readily support constructing much larger datasets whenever needed.
>
> ---
>
> ## W4. Interaction with Selective Prediction
> > *This paper would benefit from interaction with the extensive literature on selective prediction, which has the same motivation (e.g. https://aclanthology.org/2022.findings-acl.158/, https://arxiv.org/abs/2303.16857, https://arxiv.org/abs/2402.15610)*
>
> We thank the reviewer for introducing selective prediction along with several relevant references. We also agree that our framework shares the same underlying motivation as selective prediction. Accordingly, we will expand the motivation discussion and revised the related work section to incorporate connections to selective prediction literature, as shown below.
>
>  **Selective Approaches to Relevance Assessment.** Recent work on selective prediction [1, 2, 3] shows that models can reduce errors by abstaining or escalating uncertain cases rather than committing to unreliable decisions. Building on this idea, AI-human hybrid methods [4, 5], which leverage LLMs for straightforward cases and rely on human intervention for uncertain ones, have been proposed to address inaccuracies in fully automated labeling, such as collaborative annotation [6] and verification pipelines [7]. Yet, there has been limited research on their application to relevance assessment. Beyond the confidence-based method, LARA [8] calibrates LLM confidence into a relevance probability using human-labeled data to mitigate miscalibration and overconfidence. Despite these advances, hybrid pipelines remain limited by single-model accuracy and a heavy reliance on human data. In contrast, our approach mitigates it through opposing opinions and reciprocal critique in multi-round debate, eliminating the need for calibration and the reliance on additional training with human supervision.
>
> [1] Varshney et al., "Investigating Selective Prediction Approaches Across Several Tasks in IID, OOD, and Adversarial Settings", ACL 2022. \
> [2] Stengel-Eskin and Durme, "Did You Mean...? Confidence-based Trade-offs in Semantic Parsing", EMNLP 2023. \
> [3] Srinivasan et al., "Selective “Selective Prediction”: Reducing Unnecessary Abstention in Vision-Language Reasoning", ACL 2024.\
> [4] Xu et al., "RLTHF: Targeted Human Feedback for LLM Alignment", ICML 2025.\
> [5] Sahitaj et al., "Hybrid Annotation for Propaganda Detection: Integrating LLM Pre-Annotations with Human Intelligence", NLP4PI 2025.\
> [6] Kim et al., "MEGAnno+: A Human-LLM Collaborative Annotation System", EACL 2024.\
> [7] Wang et al., "Human-LLM Collaborative Annotation Through Effective Verification of LLM Labels", CHI 2024.\
> [8] Takehi et al., "LLM-Assisted Relevance Assessments: When Should We Ask LLMs for Help?", SIGIR 2025.
>
> ---
>
> ## W5. Acc-Escalation Ratio Plot
> > *Comparison to LARA would be easier to understand as a plot with different confidence thresholds/escalation ratios on the x axis*
>
> We appreciate the reviewer's suggestions, which have strengthened the clarity of our work. We have included an accuray plot with different confidence thresholds/escalation ratios in Figure 2 on page 5. It is more comprehensive to present the full trade-off curve, as this provides a clearer depiction of how accuracy changes with different escalation ratios.
>
> ---

---

> ### Author Response · Authors · 2025-11-25
> **Response to Reviewer M8G8 (4/5)**
>
> ---
>
> ## Q1. Order-Dependence
> > *Order-dependence: LLMs are often prone to sycophancy and there's a documented trend of models picking whichever answer goes first. Does it matter which agent goes first? It would be good to have an ablation showing stance order doesn't matter.*
>
> We are grateful for the reviewer's thoughtful recommendation regarding the order dependence ablation study. To examine whether the debate outcome depends on which agent speaks first, we conducted an ablation where we swapped the stance order while keeping the model, prompt, temperature, and number of rounds identical.
>
> **Label Stability under Stance Order Swap**
> We compared the final debate labels for all consensus cases produced under the "Relevant-first (RF)" and "Irrelevant-first (IF)" initializations. In the table below, each row corresponds to the final labels obtained when agents start from the Relevant-first stance, and each column corresponds to the final labels obtained when agents start from the Irrelevant-first stance. The table therefore shows how consistent the final labels are across the two settings.
>
> | RF \ IF     | Irrelevant | Relevant | Total |
> | --- | --- | --- | --- |
> | **Irrelevant** | 393 | 14 | 407 |
> | **Relevant** | 8 | 261 | 269 |
> | **Total** | 401 | 275 | 676 |
>
> Across 676 queries, 96.7% (654/676) of predictions remained identical after swapping the agent order, and only 3.3% (22/676) changed. Importantly, the labels changed in both directions. 8 cases flipped from relevant to irrelevant, while 14 cases flipped from irrelevant to relevant. If the model were simply following the first agent, label changes would be heavily skewed toward one direction. The nearly balanced flip pattern instead demonstrates no detectable preference for the initial stance.
>
> **Performance Comparison**
> | Method         | Recall (Relevance) | Recall (Irrelevance) | bAcc | Escalation Ratio |
> | --- | --- | --- | --- | --- |
> | ```Relevant First (RF)``` | 98.4% | 91.9% | 95.2% | 3.8% |
> | ```Irrlevant First (IF)```   | 97.3% | 90.1% | 93.7% | 4.4% |
>
> Balanced accuracy remains nearly unchanged, indicating that overall evaluation quality is unaffected by order. This indicates that the debate outcomes are **not affected by first-speaker bias or order-dependence**.
>
> ---
>
> ## Q2. Consensus Evaluation
> >*How do you evaluate consensus? Are models instructed to format answers in an extractable way?*
> >
> We appreciate this question, as it helps clarify our consensus evaluation and formatting instructions. We specify an explicit output template in the prompt to ensure extractable predictions, using the instruction:
>
> Provide your output as a properly formatted JSON object. No additional explanation.
> ```
> {
>   "reference": ["reference sentence 1 if exists", ...],
>   "reason": "Explain your reasoning step by step and conclude with a clear justification of the final decision. (Max 100 words)",
>   "response": "yes or no"
> }
> ```
> To guarantee extractability, we either parse the model output directly as JSON or apply regex-based post-processing to recover the intended JSON structure.
> For consensus evaluation, we define agreement as both agents producing the same `response` value (`yes` or `no`); otherwise, the pair is counted as disagreement.
>
> ---

---

> ### Author Response · Authors · 2025-11-25
> **Response to Reviewer M8G8 (5/5)**
>
> ---
>
> ## Q3. What was the reason for human disagreement on human eval?
> > *What was the reason for human disagreement on human eval?*
>
> Thank you for the question, as this point helps clarify the underlying sources of disagreement in human evaluation. We attribute human disagreement to **"different interpretation strategies"** when assessing whether a chunk grounds an answer. Annotators may prioritize different aspects, such as exact numerical matches versus conceptual alignment, or explicit statements versus implied information. Here is an example from `BRIDGE` that showed disagreement:
>
> ### Case Example
> | Element | Content |
> |---------|---------|
> | **Query** | how much do tooth veneers cost |
> | **GT Answer** | \\$800 to \\$2000 per tooth |
> | **Chunk** | composite veneers cost per tooth ranges between \\$250 and \\$1500 porcelain veneers cost per tooth ranges between \\$925 and \\$2500 compare with the cost of mini implants the cost of dentures and the cost of tooth bridges |
> | **Agent 1 Reasoning (Supporting)** | I evaluated the answer against the document's content, considering scope, intent, and direct information. The document provides specific cost ranges for veneers, which match the answer. |
> | **Agent 2 Reasoning (Unsupporting)** | I evaluated the answer and found the document provides categorically distinct price ranges that do not explicitly match the answer's single range, requiring implications beyond what is directly stated.|
> | **Human Annotations** | 1 Relevant, 2 Irrelevant |
>
>
> **Analysis**: The chunk distinguishes composite (\\$250-\\$1500) and porcelain (\\$925-\\$2500) veneers, while the answer provides a single range (\\$800-\\$2000). As shown in the agent reasoning, Agent 1 accepts partial overlap as sufficient grounding, while Agent 2 requires categorical precision and complete alignment. This exemplifies how annotators apply different interpretation strategies—some accepting implied information while others requiring explicit statements.
>
> Such subjective cases are precisely what `DREAM` successfully identifies for human escalation. However, this shows that crowd-sourced annotators can disagree due to different interpretations, suggesting the need for expert annotators. This aligns with prior work showing human disagreements often reflect inherent ambiguity rather than annotation errors (Pavlick & Kwiatkowski, 2019). We believe helping annotators handle these subjective cases and reach consensus is an important area for future work.
>
> [1] Ellie Pavlick and Tom Kwiatkowski, "Inherent Disagreements in Human Textual Inferences", TACL, 2019.
>
> ---
>
> ## Q4. Distillation
> > *Have you thought about distillation? Past work, e.g. https://arxiv.org/abs/2402.01620 distills multi-agent interactions into a single model. In principle, you could distill the better classifications from the DREAM debate pipeline into a single judge model that at test time produces a single decision.*
>
> We appreciate the reviewer’s insightful suggestion regarding distillation and pointing us to MAGDi. Representing multi-agent interactions as graphs and distilling them into a single annotator is indeed an interesting and promising direction.
>
> In our setting, debating agents begin from opposing stances, and distilling this interaction could encourage a single model to incorporate both perspectives and reduce bias. Moreover, the distilled model would still need to detect uncertain cases, which could be handled through confidence scores, output consistency, or internal conflict signals. If such a model learns the opposing stances well and is trained across diverse domains, we believe it could be an effective and efficient annotator.
>
> ---
>
> If there are any aspects of our response that remain unclear or could benefit from further clarification, we would be more than happy to address them.
>
> ---

---

> > ### Comment · Reviewer_M8G8 · 2025-11-26
> > **Response to rebuttal**
> >
> > Thanks for the in-depth response -- I believe that the additional results on latency and the additional related work address will strengthen the soundness of the paper. I will maintain my positive assessment.

---

> ### Author Response · Authors · 2025-11-27
> **Response to Reviewer M8G8**
>
> We are glad to hear that our additional results are satisfactory for you. All discussed results (including additional latency experiments and expanded related work) have now been incorporated into the revised manuscript, which is available for immediate review (see the "magenta" and "orange" color-coded updates).
>
> If you feel that our revision has adequately addressed your concerns and questions, we would be grateful if you could consider reflecting this improvement in your evaluation.
>
> Thanks again for your constructive feedback. It has greatly helped us improve the clarity and quality of our paper.
>
> Sincerely, authors

---

### Official Review · Reviewer_MSma · 2025-11-01

**Soundness:** 3
**Presentation:** 3
**Contribution:** 3
**Rating:** 4
**Confidence:** 4

**Summary:**

This paper proposes DREAM, a multi‑agent debate framework for completing missing answer‑aware relevance annotations in IR benchmarks. Specifically, two LLM agents are initialized with opposing stances, debate for up to two rounds, and auto‑label a query–chunk pair upon consensus; persistent disagreements are escalated to human annotators together with the debate history to aid adjudication. DREAM targets accurate, low‑cost query–chunk relevance labeling and reports higher accuracy than single‑agent LLM‑as‑judge and confidence‑based escalation baselines.

Using DREAM, the authors construct BRIDGE, a refined benchmark built from subsets of BEIR and RobustQA that substantially reduces unlabeled relevant “holes.” BRIDGE adds 29,824 previously unlabeled relevant chunks—a 428% increase over the originally annotated 6,976 gold chunks—bringing the total to 36,800. The paper shows that such holes introduce systematic evaluation bias that can underestimate retriever effectiveness, and it introduces RAGAlign to measure retrieval–generation alignment in RAG systems. Results indicate stronger retrieval–generation alignment on BRIDGE than on the original benchmarks, and ablations on the expert‑annotated subset support the choice of two debate rounds and the utility of providing debate history for AI–human synergy.

**Strengths:**

1. The paper identifies and addresses a critical bottleneck in existing IR benchmarks: the incompleteness (i.e., holes) of current relevance-annotated datasets leading to unreliable retrieval performance evaluation of the RAG system.
2. The proposed multi-agent debate annotation pipeline is intuitive and clearly explained.
3. The constructed BRIDGE benchmark uncovers 4 times previously unlabeled relevant chunks compared to the original gold chunks with relatively low cost, which should be meaningful for the IR community.

**Weaknesses:**

1. **Homogeneous, 2-agent setting for multi-agent debate**. In DREAM, both agents are Llama-3.3-70B-Instruct with temperature=0. Although this is a standard minimal setting for multi-agent debate, what if we use more than 2 agents or heterogeneous-model (i.e., different LLMs for each agent) to increase the diversity during debating? Will this bring more accurate annotation?
2. **Agreement treated as reliability without analyzing "wrong-but-agree"**. There is discussion of persistence of agreement across debate rounds in section A.5. However, there are no analysis to quantify the rate of spurious consensus (cases where agents agree yet the consensus contradicts expert labels).
3. **RAGAlign analysis tied to a single generator configuration**. The RAGAlign@K analysis in section 5.2 only focus on Llama3.3-8B-Instruct. It is not shown whether the observed retrieval–generation alignment gains on BRIDGE hold for other LLM generators. Also, there seems to have a generator version discrepancy between Figure 3 (Llama‑3.3‑8B) and Appendix I.4 (Llama‑3.1‑8B).
4. **Will the more accurate chuck annotation help generation performance?** While RAGAlign improves, the paper does not report whether denser, more accurate chunk annotations translate into better end‑task generation metrics under the same generator and prompts.

**Questions:**

Please see weaknesses.

Another following up question considering W.1 and W.2 together:
1. If you increase debate diversity—by adding more agents, using heterogeneous base models, or allowing non‑zero temperatures—does the rate of spurious consensus decrease, and does end‑to‑end accuracy improve at comparable escalation cost?

---

> ### Author Response · Authors · 2025-11-25
> **Response to Reviewer MSma (1/5)**
>
> We sincerely appreciate the reviewer’s thoughtful and constructive feedback. We have carefully examined each concern and provide detailed responses below. We hope our rebuttal addresses the issues raised, and all discussed revisions will be incorporated into the updated manuscript.

---

> ### Author Response · Authors · 2025-11-25
> **Response to Reviewer MSma (2/5)**
>
> ---
>
> ## W1 & Q1. Debate Diversity - more agents, heterogeneous base models, non-zero temperatures (1/2)
>
> > ***W1-1:** In DREAM, both agents are Llama-3.3-70B-Instruct with temperature=0. Although this is a standard minimal setting for multi-agent debate, what if we use more than 2 agents or heterogeneous-model to increase the diversity during debating? Will this bring more accurate annotation?*
>
> Thanks for your suggestion on enhancing diversity in the debating setup, including (i) increasing the number of agents, (ii) using heterogeneous model families, and (iii) allowing non-zero temperature. We examined the annotation performance of our framework across the diverse setups to assess whether increased diversity indeed leads to improvements in labeling accuracy.
>
> In summary, increasing diversity in the debating setup does not bring improved annotation performance. Our results show that:
> - (i) More agents makes relevant agreement harder to reach, which reduces recall for relevance cases and slightly degrades overall accuracy.
> - (ii) Heterogeneous models reduce model-specific bias but still yield lower accuracy due to performance discrepancies between model families.
> - (iii) Higher temperatures produce long rationales, causing the debate to drift toward irrelevant decision and ultimately degrading labeling accuracy.
>
> In detail, we conducted an experiment on 728 samples using the same setup as Section 3.2.1 while varying the diversity configurations.
>
> **(i) Experiment on Number of Agents**
>
> We extend the number of agent from two to four:
> - (1) 2-Llama: Two Llama3.3-70B-Instruct (default)
> - (2) 4-Llama: Four Llama3.3-70B-Instruct
> - (3) 2-Llama & 2-Qwen: Two Llama3.3-70B-Instruct models and two Qwen2.5-72B-Instruct models), with each model family initialized with both the relevant and irrelevant stances
>
> | Method | Recall (Irrelevance) | Recall (Relevance) |  bAcc  | Escalation Ratio|
> | -------------- | ----------------------------- | ------ | ------ | ---------------- |
> | `2-Llama` | 91.9% | 98.4% | 95.2% | 3.5% |
> | `4-Llama`  | 97.2% | 91.4% | 94.3% | 11.2% |
> | `2-Llama & 2-Qwen` | 97.3% | 92.0% |94.7% | 13.0% |
>
> Increasing the number of debate models naturally lowers the consensus rate (100% − escalation ratio), as more models must agree before a final label can be assigned. Although this stronger agreement requirement could be expected to improve labeling accuracy (bAcc) on non-escalated cases, we find that bAcc actually decreases. As more models participate, the debating process becomes increasingly conservative toward relevant cases, making agreement on them harder to reach compared to irrelevant ones. This leads to a higher number of disagreements on relevant examples, which in turn reduces recall (relevance) and overall labeling accuracy.
>
> **(ii) Experiment on Model Families**
>
> We consider three more configurations:
> - (1) 2-Llama (default)
> - (2) 2-Qwen: Two Qwen2.5-72B-Instruct
> - (3) Llama(Relevant) & Qwen(Irrelevant): Llama initialized with relevant stance while Qwen with irrelevant stance
> - (4) Qwen(Relevant) & Llama(Irrelevant): The reverse of (3).
>
> | Method  | Recall (Irrelevance) | Recall (Relevance) | bAcc | Escalation Ratio |
> | -----  | ----- | ----- | ----- | ----- |
> | `2-Llama` | 91.9% | 98.4% | 95.2% | 3.5% |
> | `2-Qwen`  | 95.1% | 82.8% | 89.0% | 2.6% |
> | `Llama(Relevant) & Qwen(Irrelevant)` | 95.8% | 88.7% | 92.3% | 6.6% |
> | `Qwen(Relevant) & Llama(Irrelevant)` | 96.0% | 91.4% | 93.7% | 6.5% |
>
> The 2-Llama baseline shows higher recall (relevance), while the 2-Qwen baseline shows higher recall (irrelevance), indicating that Llama leans toward predicting "relevant" and Qwen toward "irrelevant." The heterogeneous setup reduces these biases, achieving higher  recall (irrelevance) than 2-Llama and higher recall (relevance) than 2-Qwen. However, its bAcc remains lower than 2-Llama because Qwen’s weaker performance drags down overall accuracy. This suggests that heterogeneous debate is beneficial only when all participating models are individually strong.
>
> **(iii) Experiment on Model Temperature**
>
> We adjust the temperature under `2-Llama` setup.
>
> | Temp | Recall (Irrelevance) | Recall (Relevance) | bAcc | Escalation Ratio|
> | ------ |  ----------------------------- | ------ | ------ | ---------------- |
> | ``0.0``  | 91.9% | 98.4% | 95.2% | 3.5% |
> | ``0.6``  | 93.5% | 92.7% | 93.1% | 3.8% |
> | ``1.0``  | 93.3% | 91.9% | 92.6% | 3.1% |
>
> Under higher temperatures, models produce often longer reasoning because the sampling distribution becomes flatter, making low-probability tokens more likely to be selected. This lengthening causes the debate to drift toward irrelevance, which in turn makes the final decision more conservative. Although recall (irrelevance) improves slightly, the substantial drop in recall (relevance) leads to an overall decline in labeling accuracy (bAcc).
>
> ---

---

> ### Author Response · Authors · 2025-11-25
> **Response to Reviewer MSma (3/5)**
>
> ---
>
> ## W1 & Q1. Debate Diversity - more agents, heterogeneous base models, non-zero temperatures (2/2)
>
> > ***Q1:** If you increase debate diversity—by adding more agents, using heterogeneous base models, or allowing non‑zero temperatures—does the rate of spurious consensus decrease, and does end‑to‑end accuracy improve at comparable escalation cost?*
>
> The table below reports the rate of spurious consensus (wrong-but-agree cases) across the three configurations we tested along with their annotation accuracy (bAcc): (i) increasing the number of agents, (ii) using heterogeneous model families, and (iii) allowing non-zero temperature (refer to the response to W1 for setup details). Spurious consensus is defined as the proportion of incorrect agreements among all agreement cases.
>
> |                          | 2-Llama (Default) | 2-Llama | 2-Llama | 2-Qwen | Llama & Qwen | Qwen & Llama | 4-Llama | 2-Llama & 2-Qwen |
> |--------------------------|---------|---------|---------|--------|--------------|--------------|---------|------------------|
> | # Agents                  | 2       | 2       | 2       | 2      | 2            | 2            | 4       | 4                |
> | Temperature              | 0.0     | 0.6     | 1.0     | 0.0    | 0.0          | 0.0          | 0.0     | 0.0              |
> | Spurious Concensus Ratio |   5.9%  | 6.6%    | 7.1%    | 9.3%   | 6.8%         | 5.6%         | 4.5%    | 4.7%             |
> | bAcc                     | 95.2%    | 93.1%    | 92.6%    | 89.0%   | 92.3%         | 93.7%         | 94.3%   | 94.7%            |
>
> **Enhancing diversity does decrease the rate of spurious consensus?**
>
> Although all three directions, (i) increasing the number of agents, (ii) using heterogeneous model families, and (iii) allowing non-zero temperature, are intended to increase diversity, only the "(i) increasing the number of agents" configuration reduces spurious consensus compared to the 2-Llama baseline (default setup) from 5.9% to 4.5%. In contrast, both heterogeneous models and higher temperatures slightly increase the rate of spurious consensus. This shows that diversity does not consistently mitigate incorrect agreements, and certain forms of diversity introduce additional noise that can even amplify wrong-but-agree cases.
>
> **Does end‑to‑end accuracy improve at comparable escalation cost with higher diversity?**
>
> We observe a drop in the rate of spurious consensus only when increasing the number of agents from two to four. However, the end-to-end labeling accuracy (bAcc) does not improve; instead, it slightly decreases from 95.2% to 94.7%. As discussed earlier, when more models participate, reaching agreement on relevant cases becomes harder than reaching agreement on irrelevant ones, which reduces recall (relevance) and leads to a lower overall bAcc even though spurious consensus decreases.
>
> ---

---

> ### Author Response · Authors · 2025-11-25
> **Response to Reviewer MSma (5/5)**
>
> ---
>
> ## W3. RAGAlign analysis tied to a single generator configuration
>
> > ***W3-1:** There seems to have a generator version discrepancy between Figure 3 (Llama‑3.3‑8B) and Appendix I.4 (Llama‑3.1‑8B).*
>
> The generator name in Figure 3 was a typo, and we confirm that all main experiments used Llama-3.1-8B-Instruct for generation.
>
> > ***W3-2:** The RAGAlign@K analysis in section 5.2 only focus on Llama3.3-8B-Instruct. It is not shown whether the observed retrieval–generation alignment gains on BRIDGE hold for other LLM generators.*
>
> To address the concern about generality, we additionally evaluate retrieval–generation alignment rate on two other LLM generators, Qwen2.5-7B-Instruct and Gemma-2-9b-it.
> * Qwen2.5-7B-Instruct
>
> | RAGAlign@10 | MS | NQ   | Life | Rec | Sci | Tech | Writ | Avg.     |
> | -------- | ------- | ---- | --------- | ---------- | ------- | ---------- | ------- | ------------ |
> | **Original** | 0.58    | 0.71 | 0.67      | 0.71       | 0.66    | 0.66       | 0.69    | 0.66 |
> | **BRIDGE**   | 0.83    | 0.8  | 0.71      | 0.75       | 0.75    | 0.76       | 0.73    | 0.76 |
>
>
> * Gemma-2-9b-it
>
> | RAGAlign@10 | MS   | NQ   | Life | Rec  | Sci  | Tech | Writ | Avg.     |
> | -------- | ---- | ---- | ---- | ---- | ---- | ---- | ---- | ------------ |
> | **Original** | 0.6  | 0.74 | 0.73 | 0.73 | 0.68 | 0.68 | 0.72 | 0.69 |
> | **BRIDGE**   | 0.85 | 0.86 | 0.76 | 0.78 | 0.76 | 0.76 | 0.75 | 0.78 |
>
> Both models exhibit the similar trend as Llama-3.1-8B-Instruct, demonstrating that the observed gains are not tied to a single generator configuration. We will include these results in the revised version.
>
> ---
>
> ## W4. Will the more accurate chuck annotation help generation performance?
>
> The goal of our work is not to enhance the performance of either the retriever or the generator. We focus on refining the existing IR datasets that currently provide inaccurate evaluations of retrievers, thereby improving the performance alignment between retrieval and generation in RAG. However, we agree that the reviewer’s question touches on an important and meaningful direction for extending our contributions.
>
> > ***W4-1:** Will the more accurate chuck annotation help generation performance?*
>
> Accurate chunk annotations produced by our framework are not only useful for evaluation but also highly valuable as training data. Because the labels precisely capture true relevance at the chunk level, they can directly improve downstream RAG systems in two ways. First, they enable more effective training of the base retriever: supervised contrastive training on high-quality positive/negative chunk pairs leads to stronger relevance discrimination and better recall. Second, they provide clean supervision for re-rankers, which rely heavily on fine-grained relevance signals; accurate labels allow re-rankers to learn subtle distinctions between borderline relevant versus irrelevant chunks. Together, these two benefits can substantially enhance retrieval quality, which in turn leads to more accurate and faithful generation in RAG pipelines.
>
> > ***W4-2:** While RAGAlign improves, the paper does not report whether denser, more accurate chunk annotations translate into better end‑task generation metrics under the same generator and prompts.*
>
> We are sorry for the confusion. Our work does not aim to improve RAG generation performance; instead, the goal is to build a reliable benchmark with accurate relevance annotations. More clearly, RAGAlign improvement does not indicate an improvement in either retrieval or generation performance. Instead, it addresses the misalignment between retrieval and generation evaluation. Inaccurate retrieval evaluation in existing benchmark creates this misalignment. For example, when relevant chunks are unlabeled (“holes”), correct retrieval is mistakenly counted as failure, and correct generation is misattributed to parametric knowledge rather than retrieved evidence.
>
> To solve this issue, we annotate missing relevant chunks to enable reliable retriever evaluation, resulting in retrieval-generation alignment improvement. This leads to a more accurate and interpertable analysis of the interaction between the retriever and generator in RAG systems.
>
> ---
>
> If there are any aspects of our response that remain unclear or could benefit from further clarification, we would be more than happy to address them.
>
> ---

---

> ### Author Response · Authors · 2025-11-25
> **Response to Reviewer MSma (4/5)**
>
> ---
>
> ## W2. Agreement treated as reliability without analyzing "wrong-but-agree" across debate round
> > *Agreement treated as reliability without analyzing wrong-but-agree. There is discussion of persistence of agreement across debate rounds in section A.5. However, there are no analysis to quantify the rate of spurious consensus*
>
> We consider this a very good follow-up question to W1 and Q1, asking whether increasing the number of debate rounds can reduce the spurious consensus ratio. Thus, we extend the results in Section A.5 to quantify the spurious consensus rate across debate rounds.
>
> **The Rate of Suprious Consensus over Multiple Rounds**
>
> We add the spurious consensus ratio to the table in Appendix A.5, which reports the persistence-of-agreement study across debate rounds.
>
> | Round                           | R1    | R2    | R3    | R4    | R5    |
> | ------------------------------- | ----- | ----- | ----- | ----- | ----- |
> | Spurious Consensus Ratio (%) | 5.0 | 3.9 | 5.0 | 5.1 | 5.1 |
> | Recall (Irrelevance) | 92.7% | 94.5% | 94.0% | 94.0% | 94.0% |
> | Recall (Relevance) | 99.0% | 98.6% | 96.4% | 96.1% | 96.1% |
> | bAcc | 95.8% | 96.5% | 95.2% | 95.0% | 95.1% |
>
> Increasing the number of debate rounds does not consistently reduce the spurious consensus rate. The lowest rate appears at R2 (3.9%), after which it rises again, reaching 5.1% at R5. That is, more rounds eventually introduce additional noise into the discussion, making incorrect agreements more likely rather than less.
>
> In detail, from R1 to R2, we observe cases where spurious consensuses or initial disagreements are corrected in R2, because these errors are relatively easy for LLMs to reassess once the opposing arguments are explicitly surfaced. However, extending the debate beyond two rounds pushes the cases that remain ambiguous even after R2 toward an incorrect "irrelevance" consensus, as evidenced by the drop in recall (relevance) while irrelevance recall remains stable.  These results demonstrate that using more than two rounds carries a risk of amplifying spurious consensus.
>
> Refer to the qualitative examples below.
>
> * **Example: Correct Consensus (R1 &rarr; R2)**
> Some cases that begin with disagreement in R1 transition to consensus in R2. In these cases, agents actively review and respond to each other's arguments, and the revised stance in R2 often corrects an initial mistake with deeper reasoning.
>     ```
>     <R1>-Disagreement
>     - AgentA(Relevant): "The document provides a specific example of calculating boat transport cost, which aligns with the answer's scope. It directly provides a cost per mile, consistent with the answer."
>     - AgentB(Irrelevant): "The document provides a specific example, but doesn't directly state the general cost range. It implies a calculation method, not a direct answer."
>
>     <R2>-Consensus
>     - AgentA(Irrelevant): "I interpreted the document as providing a specific example of calculating boat transport cost. - AgentB argued it doesn't directly state the general cost range, but I think the example implies a range. However, guideline 3 states the document must directly provide the answer without implications."
>     - AgentB(Irrelevant): "I interpreted the document as providing a specific example, not a general cost range. AgentA's position is incorrect as it implies a calculation method, not a direct answer. The document does not directly state the general cost range."
>
> * **Example: Amplified Spurious Consensus After R2**
> Ambiguous cases tend to converge to an incorrect irrelevant consensus when additional debate rounds are forced. Continuing the debate risks pushing these ambiguous instances toward spurious consensus.
>     ```
>     <R1>-Disagreement: (...)
>
>     <R2>-Disagreement
>     - AgentA(Relevant): "The document explains mutual exclusivity and independence, implying that mutually exclusive events cannot be independent, supporting answers [0], [1], and [2]."
>     - AgentB(Irrelevant): "The document explains mutual exclusivity and independence, implying that mutually exclusive events cannot be independent, but does not directly state this without implication."
>
>     <R3>-Spurious Consensus
>     - AgentA(Irrelevant): "The document explains mutual exclusivity and independence, but does not directly state that mutually exclusive events cannot be independent without implication, thus not fully justifying answers."
>     - AgentB(Irrelevant): "The document explains mutual exclusivity and independence but doesn't directly state that mutually exclusive events cannot be independent without implication."
>
>     <R4>-Spurious Consensus
>     - AgentA(Irrelevant): "The document explains mutual exclusivity and independence but doesn't directly state that mutually exclusive events cannot be independent without implication, thus not fully justifying answers."
>     - AgentB(Irrelevant): "Document explains mutual exclusivity and independence but doesn't directly state their relationship, thus not fully justifying answers."
>
> ---

---

> ### Author Response · Authors · 2025-11-27
>
> We hope that our clarifications and additional experiments provided above have addressed your concerns. If you feel that our revision has adequately addressed your concerns and questions, we would be grateful if you could consider reflecting this improvement in your evaluation (see the  "orange" and "cyan" and color-coded updates in the revised manuscript). Please feel free to let us know if any points remain unclear.
>
> Thanks again for your constructive feedback. It has greatly helped us improve the clarity and quality of our paper.
>
> Sincerely, authors

---

### Official Review · Reviewer_z9oR · 2025-11-02

**Soundness:** 2
**Presentation:** 2
**Contribution:** 2
**Rating:** 4
**Confidence:** 3

**Summary:**

The paper proposes DREAM, a debate-based framework for completing missing annotations in information retrieval (IR) benchmarks. It introduces a multi-agent relevance assessment process where two LLMs with opposing stances critique each other to reach agreement, using disagreement as a natural signal for human escalation. Building on DREAM, the authors create BRIDGE, a refined IR benchmark that fills missing relevance labels and enables fairer evaluation of retrieval and retrieval-augmented generation (RAG) systems.

**Strengths:**

- The paper is well-written and easy to follow
- The method itself is intuitive and fitting for the problem itself and seems to outperform baselines and work well empirically.

**Weaknesses:**

- Could also compare with this work and related works which use multi-agent debate to improve performance of RAG systems and contrast with these: https://arxiv.org/abs/2504.13079, https://arxiv.org/abs/2501.00332
- Could clarify on how the quality of the benchmark is impacted by the choice of number of agents and which model families these models come from.

**Questions:**

See above

---

> ### Author Response · Authors · 2025-11-25
> **Response to Reviewer z9oR (1/2)**
>
> ---
> We sincerely appreciate for the reveiwer's thoughtful comments. We have carefully reviewed your comments and provide detailed responses to each point below. We hope our rebuttal adequately addresses the raised concerns, and all discussed revisions will be incorporated into the updated manuscript.
>
> ---
> ## W1. Related work on Multi-agent Debate
> > *Could also compare with this work and related works which use multi-agent debate to improve performance of RAG systems and contrast with these: https://arxiv.org/abs/2504.13079, https://arxiv.org/abs/2501.00332*
>
> As you suggested, we further reviewed related work on multi-agent debate in RAG and general LLM tasks, and analyzed their task settings and methodological designs (see the table below). However, none of these works tackle relevance assessment in IR; existing approaches primarily focus on AI–AI collaboration to improve model performance, such as refining answers, resolving inconsistencies, or enhancing response quality.
>
> The core distinction of our work is that we adapt multi-agent debate to AI–Human collaborative relevance assessment: our framework uses debate signals to generate high-quality labels, identify uncertain cases that should be escalated to humans, and provide debate histories that enhance human judgment.
>
>
> | Paper | Goal| Task | Method |
> |---------|---------|---------|---------|
> |Wang et al., 2024 [8]| Improve performance of RAG systems | Handle interdocument conflicts, misinformation, and noise in retrieved content| Document-level agents debate with aggregator for final answer |
> |Chang et al., 2024 [1]| Improve performance of RAG systems | Improve response quality in RAG over the low quality of retrieval documents | Sequential multi-agent pipeline: generation → relevance filtering → refined answer |
> |Du et al., 2023 [5]| Improve performance of LLM in general generation| General text genertaion and reasoning enhancement | Multi-round debate between two agents |
> |Chen et al., 2024 [6]| Improve performance of LLM in general generation| General text genertaion and reasoning enhancement | Multi-agent debate across model families with confidence scoring until consensus  |
> |Chan et al., 2023 [9]| Improve evaluation quality |Assess generated responses of LLM | Multi-agent discussion with different ordering strategies (sequential, simultaneous, with/without summarizer) |
> |DREAM (Ours) | Improve annotation quality |Assess relevance between query and chunk | Adversarial debate with escalation: consensus cases auto-labeled, disagreement cases sent to human annotators |
>
> Based on this analysis, we will strengthen the Related Work section and clarify how our framework differs from prior approaches, as summarized below:
>
> **Multi-agent Debate.** Over a single-agent setup, recent studies have explored utilizing multiple agents with distinct roles that collaborate [1, 2]. In particular, multi-agent debate has been widely examined, where agents share their answers and rationales, critique one another, and iteratively refine their outputs over multiple rounds [3, 4]. Such debate-style collaboration has been applied to improve LLM and RAG answer quality [5, 6, 7], to handle knowledge conflicts in RAG [8], and to enhance response evaluation [9] and data annotation [10]. While existing work focuses primarily on AI-AI collaboration, we instead leverage multi-agent debate on AI-Human collaboration for IR relevance assessment. Our framework uses multi-agent debate not only to produce high-quality annotations, but also to determine when to escalate uncertain cases to humans and to improve human judgments through providing debate histories.
>
> [1] Chang et al., "MAIN-RAG: Multi-Agent Filtering Retrieval-Augmented Generation", ACL 2024.\
> [2] Shen et al., "Small LLMs Are Weak Tool Learners: A Multi-LLM Agent", EMNLP 2024.\
> [3] Xiong et al., "Examining Inter-Consistency of Large Language Models Collaboration: An In-depth Analysis via Debate", EMNLP 2023.\
> [4] Liang et al., "Encouraging Divergent Thinking in Large Language Models through Multi-Agent Debate", EMNLP 2024.\
> [5] Du et al., "Improving Factuality and Reasoning in Language Models through Multiagent Debate", ICML 2023.\
> [6] Chen et al., "ReConcile: Round-Table Conference Improves Reasoning via Consensus among Diverse LLMs", ACL 2024.\
> [7] Khan et al., "Debating with More Persuasive LLMs Leads to More Truthful Answers", ICML 2024.\
> [8] Wang et al., "Retrieval-Augmented Generation with Conflicting Evidence", COLM 2025.\
> [9] Chan et al., "ChatEval: Towards Better LLM-based Evaluators through Multi-Agent Debate", ICLR 2023.\
> [10] Tseng et al., "Evaluating Large Language Models as Expert Annotators", COLM 2025.
>
> ---

---

> ### Author Response · Authors · 2025-11-25
> **Response to Reviewer z9oR (2/2)**
>
> ---
>
> ## W2. Number of Agents and Model Families
> > *Could clarify on how the quality of the benchmark is impacted by the choice of number of agents and which model families these models come from.*
>
> Thanks for the suggestion. We examined the annotation performance of our framework across diverse multi-agent debate settings, as stronger annotation performance directly contributes to constructing higher-quality benchmarks.
>
> In summary, our results confirm that: (i) adding more agents leads to more conservative relevant consensus, slighltly degrading annotation performance; and (ii) although utilizing heterogeneous models (i.e., Qwen2.5-72B-Instruct and Llama3.3-70B-Instruct) mitigates the bias present in homogeneous setups, the overall annotation performance decreases compared to Llama-only setup due to Qwen's weaker performance.
>
> In detail, we ran a experiment on 728 samples using same setup as in Section 3.2.1, varying (i) the number of agents and (ii) the model families. We use Llama3.3-70B-Instruct and Qwen2.5-72B-Instruct, and compute performance on non-escalated cases where the agents reach consensus in round 2.
>
> **(i) Experiment on Number of Agents**
>
> We extend the number of agents from two to four under different configurations:
> - (1) 2 Llama3.3-70B-Instruct (Default)
> - (2) 4 Llama3.3-70B-Instruct: two begin from the relevant stance and the other two from the irrelevant stance
> - (3) 2 Llama3.3-70B-Instruct & 2 Qwen2.5-72B-Instruct with each model family initialized with both the "relevant" and "irrelevant" stances
>
> | Method  | Recall (Irrelevance) | Recall (Relevance) | bAcc |Escalation Ratio|
> | ----- | ----- | ----- | ----- | ----- |
> | `2-Llama` | 91.9% | 98.4 | 95.2 | 3.5% |
> | `4-Llama` | 97.2% | 91.4% | 94.3% | 11.2% |
> | `2-Llama & 2-Qwen` | 97.3% | 92.0% | 94.7% | 13.0% |
>
> As more agents are added, their differing reasoning paths naturally increase reasoning diversity. However, since consensus now requires agreement among more agents, the consensus ratio drops sharply (i.e., escalation ratio increases sharply). This makes the system more conservative in assigning "relevant," resulting in lower recall (relevance) but higher recall (irrelevance). The drop in recall (relevance) is larger than the gain in recall (irrelevance), the overall annotation performance declines slightly.
>
> **(ii) Experiment on Model Families**
>
> We consider three more configurations:
> - (1) 2-Llama: Both agents use Llama3.3-70B-Instruct (our default setup)
> - (2) 2-Qwen: Both agents use Qwen2.5-72B-Instruct
> - (3) Llama(Relevant) & Qwen(Irrelevant): Agent initialized with "relevant" stance uses Llama, and agent with "irrelevant" stance uses Qwen
> - (4) Qwen(Relevant) & Llama(Irrelevant): The reverse of (3) to isolate the effect of model choice from stance assignment, since each model may perform differently depending on which stance it defends.
>
> | Method | Recall (Irrelevance) | Recall (Relevance) | bAcc | Escalation Ratio |
> | ----- | ----- | ----- | ----- | ----- |
> | `2-Llama` | 91.9% | 98.4 | 95.2 | 3.5% |
> | `2-Qwen`  | 95.1% | 82.8% | 89.0% | 2.6% |
> | `Llama(Relevant) & Qwen(Irrelevant)` | 95.8% | 88.7% | 92.3% | 6.6% |
> | `Qwen(Relevant) & Llama(Irrelevant)` | 96.0% | 91.4% | 93.7% | 6.5% |
>
> The 2-Llama baseline achieves higher relevant recall than irrelevant recall, whereas the 2-Qwen baseline shows the opposite pattern. This demostrates that Llama tends to predict "relevant" and Qwen tends to predict "irrelevant," reflecting the inherent bias of each single-model family. The heterogeneous setup mitigates the bias present in homogeneous setups, as shown by higher irrelevant recall than the 2-Llama baseline and higher relevant recall than the 2-Qwen baseline. However, despite reducing bias, the heterogeneous setup yields lower bAcc than the 2-Llama due to Qwen's weaker overall performance. This suggests that heterogeneous debate becomes effective only when all participating models are individually strong.
>
> In addition, the greater diversity introduced by heterogeneous model families also reduces the consensus ratio (i.e., increases the escalation ratio), similar to the effect of adding more agents to the debate.
>
> ---

---

> ### Author Response · Authors · 2025-11-27
>
> We hope that our clarifications and additional experiments provided above have addressed your concerns. If you feel that our revision has adequately addressed your concerns and questions, we would be grateful if you could consider reflecting this improvement in your evaluation (see the "orange" and "olive" color-coded updates in the revised manuscript). Please feel free to let us know if any points remain unclear.
>
> Thanks again for your constructive feedback. It has greatly helped us improve the clarity and quality of our paper.
>
> Sincerely, authors

---

### Meta-Review · Area_Chair_eRNG · 2026-01-07

**Summary:**

This paper’s main contribution is an empirical evaluation of a simple multi-round debate methodology to address missing relevant documents in information retrieval benchmarks. This is a significant issue for IR and RAG system evaluation and therefore applying the most advanced methods to this problem is a significant, albeit minor, contribution to the community. In order to demonstrate their contribution, the authors present an IR benchmark made by using their framework to refine several existing benchmarks. They find that retriever rankings change after the refinement and the alignment between retrieval and generation metrics in RAG systems increase. The empirical work is well-presented and sound.

Given that the two most thorough reviewers (R3 and R4) kept their positive scores and all the weaknesses pointed out by the other two reviewers were addressed adequately, I will recommend that this paper be accepted.

- The only concern that was only partially addressed was that no other multi-agent debate designs were tested, they were just discussed. This lowers the comprehensiveness of this work but does not take away from its practical contributions to improving IR system evaluation.

**Reviewer Concerns:**

R1: 4,3
- Comparison with other multi-agent work
	- The authors carry out a thorough review of prior multi-agent work and claim that these debate strategies are not applicable to their setting since they are not applied to evaluating annotations. This is not a very convincing argument since the debate design proposed by the authors (opposing views reaching a consensus or not) is just one potential design that could be benchmarked against similar strategies for completeness.
- How does quality vary with # of agents and model families?
	- The added analysis on # of agents and different agent families is comprehensive. It shows that increasing the number of agents and adding other families both increase the difficulty for reaching consensus on a “Relevant” label which leads to lower Relevance recall and higher escalation rates. This concern is well addressed.

R2: 4,4
- Effects of homogeneous agents in debate
	- Same as R1 response to similar question.
- “Wrong-but-Agree” analysis missing
	- The authors include a quantitative analysis on the “wrong-but-agree” question across debate rounds and find that this “spurious consensus problem” gets worse beyond the second round of debate. This is also shown with an example of how spurious consensus gets more entrenched when more rounds are forced.
	-  The authors also present the spurious consensus percentages for the new diversity experiments (more agents/different models/higher temperatures). I did not see it added to the paper but I think it should be added.
	- The spurious consensus metric also seems important enough to be displayed more prominently throughout the paper, perhaps as an additional metric.
- RAG analysis only tied to one generator
	- The same analysis was done with Qwen2.5-7B-Instruct and Gemma-2-9b-it and the similar trends were found.
- Do more accurate chunk annotations help performance in RAG?
	- More accurate chunk-level annotations can help us train better retrievers and evaluate the ones we have more accurately. These improvements are likely to propagate to RAG systems but this is outside this paper’s scope, I tend to agree.

R3: 6,4 (Reviewer kept positive score)
- Latency and costs should be discussed
	- Additional analysis was added to paper.
- Related works on multi-agent debates are missing from discussion
	- Similar discussion to the one for R1’s concern. This seems to have been accepted by reviewer as sufficient for acceptance.
- Dataset is quite small
	- The idea was to build a proof of concept with DREAM rather than build a large dataset but scaling is cheap and simple.
- Relation to selective prediction
	- Added discussion in Related Works
- Order dependence
	- Debate outcomes are minimally affected by first-speaker
- Why was there human disagreement on human evaluation?
	- Authors claim that disagreement comes from inherent ambiguity.

R4: 6,5 (Reviewer kept positive score)
- Multi-hop scenario
	- Query decomposition could allow for DREAM to address some multi-hop settings.
- Do further rounds improve performance?
	- Same experiment as “Wrong but agree” in R2 analysis where agents were forced to continue debate over 5 rounds. Two rounds appears to be optimal in this setting.
- Effect of more diverse agents?
	- Same as R1 response for similar question

**Reviewer Scores:**

- z9oR 4 -> 4
	- Reviewer would have likely kept their lower score due to overall negative perspective on the paper (all “fair” sub-scores)
- MSma 4 -> 6
	- After a thorough rebuttal, this reviewer would have likely raised their score given their overall positive view of the paper (all “good” sub-scores)
- M8G8 6 -> 6
	- Reviewer kept score
- 58tj 6 -> 6
	- Reviewer kept score

---

### Decision · Program_Chairs · 2026-01-26

Accept (Poster)